



# The Environment and Climate Change Canada Carbon Assimilation System (EC-CAS v1.0) : demonstration with simulated CO observations

Vikram Khade[1,2], Saroja M. Polavarapu[1], Michael Neish[1], Pieter L. Houtekamer[1], Dylan B.A. Jones[2], Seung-Jong Baek[1], Tailong He[2], and Sylvie Gravel[1]

[1]Environment and Climate Change Canada, 4905 Dufferin Street, Toronto, Canada, M3H 5T4
[2]Department of Physics, University of Toronto, 60 St. George Street, Toronto, Canada, M5S 1A7

**Correspondence:** Vikram Khade (vikram.khade@canada.ca)

**Abstract.** In this study, we present the development of a new coupled weather and greenhouse gas (GHG) data assimilation system based on Environment and Climate Change Canada's (ECCC's) operational Ensemble Kalman Filter (EnKF). The estimated meteorological state is augmented to include three chemical constituents: $CO_2$, CO and $CH_4$. Variable localization is used to prevent the direct update of meteorology by the observations of the constituents and vice versa. Physical localization is used to damp spurious analysis increments far from a given observation. Perturbed flux fields are used to account for the uncertainty in CO due to error in the fluxes. The system is demonstrated for the estimation of 3-dimensional CO states using simulated observations from a variety of networks. First, a hypothetically dense uniformly distributed observation network is used to demonstrate that the system is working. More realistic observation networks based on surface hourly observations, and space-based observations provide a demonstration of the complementarity of the different networks and further confirm the reasonable behaviour of the coupled assimilation system. Having demonstrated the ability to estimate CO distributions, this system will be extended to estimate surface fluxes in the future.

## 1 Introduction

Environment and Climate Change Canada (ECCC) operates a GHG measurement network which has seen rapid expansion during the past decade. ECCC also possesses a GHG inventory reporting division. As required by United Nations Framework





on Climate Change (UNFCCC) commitments, Canadian emissions are quantified and reported using bottom-up methods (NIR
20    2019, http://www.publications.gc.ca/site/eng/9.506002/publication.html). In order to assess the national impact of mitigation
efforts, knowledge of the natural sources and sinks is also needed. The challenge is that there are huge uncertainties in the
natural carbon budget for Canada. For example, Crowell et al. (2019) find a range of uncertainty estimates of Boreal North
American (which is primarily Canada plus Alaska) fluxes from 480 to 700 TgC yr$^{-1}$ for an ensemble of inversion results for
2015 or 2016. This uncertainty in the biospheric uptake is comparable to the NIR estimate of anthropogenic emissions (568
and 559 TgC yr$^{-1}$ for 2015 and 2016 respectively) for Canada. In addition, there is much unknown about the fate of carbon
stored in the permafrost under a warming climate (Voigt et al., 2019), and this will have implications for the global as well
as the Canadian carbon budget. Thus, ECCC has a need to better understand and quantify GHG sources and sinks on the
national scale. The ECCC Carbon Assimilation System (EC-CAS) was proposed to address these needs using the available
tools, namely, operational atmospheric modelling and assimilation systems. The goal of EC-CAS is to characterize $CO_2$, CO
and $CH_4$ distributions and fluxes both globally and over Canada with a focus on the natural carbon cycle. An important aspect
of the impact of climate change on boreal forests is the influence of wildfires on the carbon balance in these regions. Over the
past several decades there has been an increase in the frequency of wildfires and this trend is expected to continue (Abatzoglou
and Williams, 2016; Flannigan et al., 2009), which will have a significant impact on the Canadian carbon budget and on the
Canadian economy. It is for this reason that EC-CAS also includes CO alongside the greenhouse gases $CO_2$ and $CH_4$.

Carbon Monoxide (CO) plays a role in both tropospheric chemistry and in climate. In terms of air quality, CO is an im-
portant precursor of tropospheric ozone, but it is also a by-product of incomplete combustion and thus correlates well with
anthropogenic sources of greenhouse gases from fossil fuel and biofuel burning and from forest fires. CO has a lifetime of
1-2 months which is in-between the weather and climate timescales and thus data assimilation systems (DAS) that assimilate
CO can focus on either the air quality or the climate problem. Tropospheric pollution prediction concerns short time scales
(forecasts up to 5 days) whereas climate problems concern the estimation of surface fluxes over months to years. Those data
assimilation systems whose primary objective is to better understand and predict tropospheric pollution typically use a coupled
weather and chemistry model with short assimilation windows (e.g. 12 h) to initiate short forecasts. The CO observations are
used to estimate CO initial states for the forecasts with either an Ensemble Kalman Filter (EnKF) (Barré et al., 2015; Miyazaki
et al., 2012) or a 4-d Variational (4D-Var) approach (Inness et al., 2019, 2015). The chemistry model typically includes the
numerous gas phase and aerosol reactions relevant for air quality. On the other hand, systems focused on CO's influence on cli-
mate are typically "inversion systems" wherein observations of CO concentrations are used to estimate CO surface fluxes. Here
again, both ensemble (Miyazaki et al., 2015, 2012) and variational (Jiang et al., 2017, 2015a, b, 2013, 2011; Fortems-Cheiney
et al., 2011) approaches have been used. Typically a chemistry transport model (CTM) driven by offline meteorological analy-
ses is used. Simplified chemistry models with monthly hydroxide (OH) fields (Yin et al., 2015; Fortems-Cheiney et al., 2011;
Jiang et al., 2017, 2015a, b, 2013, 2011) or full tropospheric chemistry models (Miyazaki et al., 2015, 2012) may be used.

As noted above, the focus of EC-CAS is ultimately on flux estimation of GHGs thus inverse modelling approaches are appro-
priate. However, with this approach, the mismatch between model-predicted CO and observed CO is usually used to adjust



fluxes only though some systems also adjust CO initial conditions (eg. Fortems-Cheiney et al. (2011)). In reality, the mis-
match between modelled and observed CO is due to errors in surface fluxes, meteorological analyses, initial conditions, model
formulation, representativeness and measurements. Proper attribution of the model-data mismatch requires accurate character-
ization of all of these sources of errors. Also, flux estimates from inverse models are sensitive to the choice of meteorological
analyses (Jiang et al., 2011), errors in convective mass transfer (Jiang et al., 2013, 2011; Ott et al., 2011; Arellano and Hess,
2006), boundary layer mixing (Arellano and Hess, 2006), biogenic sources (Jiang et al., 2011), aggregation error (Jiang et al.,
2011) and OH climatology errors (Jiang et al., 2015a, b, 2011). The presence of these model and meteorological forcing errors
confounds the retrieved CO flux estimates. With a coupled weather and chemistry transport model, uncertainty in the meteoro-
logical analyses can be accounted for (Barré et al., 2015), and with an ensemble Kalman filter approach, model errors can be
directly simulated (Miyazaki et al., 2012) . Flux estimation with an EnKF using a weather/GHG model was demonstrated for
$CO_2$ by Liu et al. (2011) and Kang et al. (2012, 2011).

EC-CAS will adapt the operational Ensemble Kalman Filter (EnKF) (Houtekamer et al., 2014) to perform a coupled meteorol-
ogy, GHG state and GHG flux estimation using the approach of Liu et al. (2011); Kang et al. (2012, 2011). EC-CAS directly
simulates and accounts for all components of transport error (i.e. errors arising from model formulation, meteorological state,
GHG initial conditions) as well as observation and flux errors. See Polavarapu et al. (2016) for a detailed discussion of transport
errors. EC-CAS will also be able to handle the vast quantities of observations that are anticipated since currently, roughly $10^6$
observations are already assimilated every day for weather forecasts. The main drawback is that EC-CAS is computationally
expensive. In contrast, inverse modelling techniques are much more expedient especially for multi-year or decadal flux esti-
mates. However, inverse model results are known to be strongly model dependent (Chevallier et al., 2014, 2010; Houweling et
al., 2010; Law et al., 1996) due to the inability to adequately characterize transport errors. Since the EC-CAS flux estimation
approach has not yet been demonstated with operational weather forecast models, the development of EC-CAS is proceed-
ing in stages. The first stage is to assess the state estimation component since if it works it will give confidence about the
ensemble-based CO flux estimation. The second stage is to estimate the fluxes with this ensemble. Since each of the three EC-
CAS species has its own assimilation challenges, the testing of the assimilation scheme began with CO because of its shorter
lifetime of about 2 months. Thus the computational cost to test the new system could be minimized. The system of Barré et al.
(2015) and Gaubert et al. (2016) is similar to EC-CAS in utilizing an EnKF with a coupled atmospheric and chemistry-transport
model for flux estimation. Those articles demonstrate only CO state estimation (as in this work), as they too are proceeding in
stages.

In the present paper we introduce the first version of EC-CAS to demonstrate the extension of the ensemble Kalman filter
(Houtekamer et al., 2014) to estimate GHGs. This new coupled meteorological and GHG assimilation system is called EC-
CAS v1.0. To demonstrate that the system is working, 3-dimensional CO fields are estimated by assimilating observations
from four different networks. The outline of the paper is as follows. Section 2 describes the various components of EC-CAS
system. Section 3 presents the experimental design while section 4 describes the data assimilation (DA) experiments and their
results. Section 5 presents the conclusions of this work and delineates planned future developments of EC-CAS.



## 2 System description and development

Any data assimilation system consists of a model which produces the trial fields and a statistical technique which blends these trial fields with the observations. The first stage is referred to as the "forecast step" while the second one is called the "analysis step". This section describes these components for EC-CAS starting with an overview of the whole system in section 2.1. The forecast model, the EnKF and extensions for GHG and flux estimation are presented in sections 2.2, 2.3 and 2.4 respectively.

### 2.1 EC-CAS

The EC-CAS system consists of a coupled weather and GHG transport model as the forecast model and an Ensemble Kalman filter (EnKF) as the data assimilation technique. Figure 1 shows a schematic overview of EC-CAS. The model is initialized with N realizations of meteorological variables at every grid point. The 6 h ensemble forecasts are used as the trial fields at each data assimilation (DA) cycle and the spread of the ensemble about their mean defines the forecast error covariances. The EnKF is used to blend the trial fields with the observations to produce the analysis ensemble at each DA cycle. This blending uses the error in observations, the uncertainty in trial fields along with the correlations within the trial fields. These correlations are estimated by the sample correlations of the trial ensemble. The next section 2.2 describes the coupled meteorological and GHG transport model used in the forecast step while section 2.3 delineates the meteorological component of the EnKF. Section 2.4 describes the extension of the EnKF to include estimation of CO and other GHGs.

### 2.2 The forecast model

The forecast model used in EC-CAS is called GEM-MACH-GHG (Polavarapu et al., 2016; Neish et al., 2019). This model is a variant of GEM (Global Environmental Multiscale), ECCC's operational weather forecast model (Côté et al., 1998a, b; Girard et al., 2014) that was developed for the simulation of greenhouse gases. A detailed description of the GEM-MACH-GHG model is found in Polavarapu et al. (2016), so only a few salient points are mentioned here along with recent model updates. Compared to the operational global weather forecast model, GEM-MACH-GHG uses a lower resolution with $0.9°$ grid spacing in both latitude and longitude and $80$ vertical levels and the same lid of $0.1°$ hPa. The vertical coordinate is a type of hybrid terrain-following coordinate (Girard et al., 2014). The advection scheme uses a semi-Lagrangian approach for both meteorology and tracers. Modifications were implemented to conserve tracer mass on the global scale (see Polavarapu et al. (2016)). This included defining tracer variables as dry mole fractions. In addition, tracers are transported through the Kain-Fritsch deep convection scheme (Kain and Fritsch, 1990; Kain, 2004) but not through a shallow convection scheme. The boundary layer scheme uses a prognostic turbulent kinetic energy (TKE) equation to specify the thermal eddy diffusivity (see McTaggart-Cowan and Zadra (2015)). In Polavarapu et al. (2016, 2018), it was necessary to impose a minimum value of $10$ $\mathrm{m^2s^{-1}}$ in the boundary layer. However, recent model improvements enabled the minimum value to be lowered to $1\ \mathrm{m^2s^{-1}}$ as





in Kim et al. (2020). An operational air quality forecast model based on GEM is used to produce 48 h forecasts of air quality health index on a limited area domain covering most of North America. This model is called GEM-MACH (Anselmo et al., 2010; Gong et al., 2015; Pavlovic et al., 2016) and it employs moderately detailed parameterizations of tropospheric chemistry using 42 gas-phase species, 20 aqueous-phase species, and nine aerosol chemical components. In contrast, GEM-MACH-GHG

removes the tropospheric chemistry module entirely for $CO_2$ and replaces it with simple parameterized chemistry for $CH_4$ and CO. This is because GEM-MACH-GHG is used for multi-year simulations and flux estimations of long-lived constituents with an EnKF so the computational expense of complete chemistry is prohibitive and difficult to justify for a system focused on GHG fluxes. However, other model processes from GEM-MACH are used in GEM-MACH-GHG, namely, the vertical diffusion and emissions injection. In GEM-MACH-GHG, the methane ($CH_4$) parameterization involves a single loss rate with

a monthly [OH] climatology. The rate constant is specified following the JPL (2011) formulation for bimolecular reaction for methane (see their page 1-12). The parameterized chemistry model used for CO is identical to that used in GEOS-Chem (http://geos-chem.org) in that CO destruction is parameterized following JPL (2011). The same [OH] climatology is used for $CH_4$ and CO. Specifically, the [OH] monthly climatology is from Spivakovsky et al. (2000) regridded to GEM-MACH-GHG's grid. The production of CO from $CH_4$ is computed assuming each methane molecule destroyed becomes a CO molecule.

For the $CH_4$ simulation, the fluxes were obtained from CT-$CH_4$ (Bruhwiler et al., 2014). Since CT-$CH_4$ fluxes are available from 2000-2010, the last 5-year mean (2006-2010) fluxes were used as the fluxes for the 2015 EC-CAS simulation. The initial condition (IC) for $CH_4$ for 1 January 2015 was approximated with the $CH_4$ atmospheric mole fractions from CT-$CH_4$ at the end of 2010 plus a globally uniform offset to account for the increase in $CH_4$ from 2010 to 2015 (30 ppb, estimated based on the difference from observations at the South Pole). Even though the initial condition is not correct, the impact of the errors

in the $CH_4$ initial condition (the synoptic spatial patterns) dissipates within weeks. These prescribed $CH_4$ fluxes and initial conditions appear reasonable as the model simulated $CH_4$ compares well with surface observations. To define the CO initial state, an inversion constrained by space based observations from MOPITT (Measurement of Pollution in the Troposphere) instrument v7J (Drummond, 1992) was performed with GEOS-Chem on a $4° \times 5°$ grid. The CO combustion emissions are from Hemispheric Transport of Air Pollutants (http://www.htap.org) (Janssens-Maenhout et al., 2015). Biogenic emissions of

isoprene, methanol, acetone, and monoterpenes are from a GEM-MACH simulation, with an assumed yield of CO from the oxidation of these hydrocarbons that is based on the GEOS-Chem CO-only simulation employed in Kopacz et al. (2010) and Jiang et al. (2011, 2015a, 2017). The monthly CO posterior fluxes obtained for December 2014 and throughout 2015 were used in EC-CAS EnKF cycles. Since GEOS-Chem is widely used for assimilation of MOPITT CO data, we use a posterior CO distribution from GEOS-Chem for 1 December 2014 18:00:00 UTC as the initial state on 27 December 2014 18:00:00 UTC.

## 2.3 EnKF equations for meteorology

The Kalman Filter equation (Ghil et al., 1981; Cohn and Parrish, 1991) at a particular DA cycle at time $t$ is given by,

$$\mathbf{x}^a(t) \;=\; \mathbf{x}^f(t) \;+\; \mathbf{P}^f(t)\mathbf{H}^T(t)\left[\mathbf{H}(t)\mathbf{P}^f(t)\mathbf{H}^T(t)+\mathbf{R}(t)\right]^{-1}\left(\mathbf{y}^o(t)-\mathbf{H}(t)\mathbf{x}^f(t)\right) \qquad (1)$$





In this equation $\mathbf{x}^f$ and $\mathbf{x}^a$ are state vectors of dimension $d = 400 \times 200 \times (80 \times 4 + 2)$. The dimensionality of the model grid

is $400 \times 200$. The number of vertical levels is $80$. There are four 3-dimensional meteorological variables, namely temperature, two components of winds and humidity. In addition, the state vector includes the 2-dimensional fields of surface pressure and radiative temperature at the surface. $\mathbf{x}^f$ is the trial field produced by a 6 hour forecast of GEM-MACH. $\mathbf{x}^a$ is the analysis produced by combining the information content in the trial fields and the observations. $\mathbf{P}^f$ is the forecast error covariance matrix calculated using the spread of the forecast ensemble about its mean. $\mathbf{y}^o$ is the observation vector of dimension $m$ and matrix $\mathbf{R}$ of dimension $m \times m$ represents the error in $\mathbf{y}^o$. $\mathbf{H}$ is the forward operator which maps the model state to the

observation space.

The quantity,

$$\mathbf{K}(t) \quad = \quad \mathbf{P}^f \mathbf{H}^T \; [\mathbf{H}\mathbf{P}^f \mathbf{H}^T + \mathbf{R}]^{-1} \tag{2}$$

in equation 1 is known as the Kalman gain, where the $t$ in the parenthesis for quantities on the right side are dropped for readability.

$(\mathbf{H}\mathbf{P}^f\mathbf{H}^T + \mathbf{R})$ is a huge matrix (order $\approx 10^6$) (Houtekamer et al., 2019) and its inversion is computationally onerous. This problem is circumvented by solving this equation *sequentially* (Cohn and Parrish, 1991; Anderson, 2001; Houtekamer and Mitchell, 2001). In sequential processing, the total number of observations $m$ are subdivided into $N_b$ subsets, known as *batches* containing at most $p$ observations each.

Then, the assimilation proceeds as follows :

$$\mathbf{x}_1^a(t) \quad = \quad \mathbf{x}^f(t) + \mathbf{P}^f\mathbf{H}_1^T[\mathbf{H}_1\mathbf{P}^f\mathbf{H}_1^T + \mathbf{R}_1]^{-1}(\mathbf{y}_1^o - \mathbf{H}_1\mathbf{x}^f) \qquad \textit{Pass 1}$$

$$\mathbf{x}_2^a(t) \quad = \quad \mathbf{x}_1^a(t) + \mathbf{P}^f\mathbf{H}_2^T[\mathbf{H}_2\mathbf{P}^f\mathbf{H}_2^T + \mathbf{R}_2]^{-1}(\mathbf{y}_2^o - \mathbf{H}_2\mathbf{x}_1^a) \qquad \textit{Pass 2}$$

$$\vdots$$

$$\mathbf{x}_{N_b}^a(t) \quad = \quad \mathbf{x}_{N_b-1}^a(t) + \mathbf{P}^f\mathbf{H}_{N_b}^T[\mathbf{H}_{N_b}\mathbf{P}^f\mathbf{H}_{N_b}^T + \mathbf{R}_{N_b}]^{-1}(\mathbf{y}_{N_b}^o - \mathbf{H}_{N_b}\mathbf{x}_{N_b-1}^a) \qquad \textit{Pass } N_b$$

The subscripts $1, 2, ..., N_b$ represent the pass numbers. $\mathbf{x}_{N_b}^a(t)$ is the updated state (as if all the observations were processed

simultaneously). The analysis from a given pass is used as the trial field in the next pass. At each pass at most 600 observations are assimilated.





Though the covariance estimate $\mathbf{P}^f$ obtained from the ensemble is state dependent, owing to the small size of the ensemble this estimate is noisy. This is remedied by the use of physical localization. The Kalman gain in equation 2 is modified as,

$$\mathbf{K}(t) \quad = \quad (\rho_{\mathbf{m}} \circ (\mathbf{P}^f \mathbf{H}^T)) \; [\rho_{\mathbf{o}} \circ (\mathbf{H} \mathbf{P}^f \mathbf{H}^T) + \mathbf{R}]^{-1} \tag{3}$$

where $\rho_{\mathbf{m}}$ and $\rho_{\mathbf{o}}$ constitute the localization in the model space and observation space, respectively and $\circ$ denotes the Hadamard product. These matrices contain weights that smoothly decrease towards zero as the distance from the observation increases. The localization in the model space ($\rho_{\mathbf{m}}$) requires the distance between observations and model coordinates while the localization in the observation space ($\rho_{\mathbf{o}}$) requires the distance between observations (Houtekamer et al., 2016). The covariances in $\mathbf{P}^f \mathbf{H}^T$ are multiplied elementwise by $\rho_{\mathbf{m}}$. Similarly the covariances in $\mathbf{H} \mathbf{P}^f \mathbf{H}^T$ are multiplied elementwise by $\rho_{\mathbf{o}}$. The en-
semble size is typically much smaller than the dimensionality of the model. For example, in this work the ensemble size is $64$ while the model dimensionality is $\approx 10^7$. Consequently, the correlation estimate calculated from the ensemble can be spurious. Localization is designed to ameliorate this problem of spurious correlations (Hamill et al., 2001). The rate of decrease of the weight is dictated by the Gaspari-Cohn function (Gaspari and Cohn, 1999; Houtekamer and Mitchell, 2001).

## 2.4 EnKF extensions for GHG and fluxes

The state vector discussed in section 2.3 is augmented to include the CO, $CO_2$, $CH_4$ fields and their fluxes. This state is referred to as the augmented state. Variable localization (Kang et al., 2011) is implemented in the EnKF code by modifying equation 3 as follows.

$$\mathbf{K}(t) \quad = \quad (\rho_{\mathbf{m}}^{\mathbf{v}} \circ \rho_{\mathbf{m}} \circ (\mathbf{P}^f \mathbf{H}^T)) \; [\rho_{\mathbf{o}}^{\mathbf{v}} \circ \rho_{\mathbf{o}} \circ (\mathbf{H} \mathbf{P}^f \mathbf{H}^T) + \mathbf{R}]^{-1} \tag{4}$$

Each element of $\rho_{\mathbf{m}}^{\mathbf{v}}$ and $\rho_{\mathbf{o}}^{\mathbf{v}}$ is either $1$ or $0$. Unlike the physical localization matrices the elements of variable localization
matrices are not distant dependent; they are rather *variable type* dependent. A given element is 1 when the row and column variable is of the same type and 0 otherwise. The $(i, j)$ th element of $\rho_{\mathbf{m}}^{\mathbf{v}}$ and $\rho_{\mathbf{o}}^{\mathbf{v}}$ is set to one if one desires an observation of the $j$th variable to impact the update of $i$th variable. Setting the $(i, j)$ th element to zero ensures that the observation of the $j$th variable does not contribute to the update of the $i$th variable. For example when the both the row and column of $\mathbf{H} \mathbf{P}^f \mathbf{H}^T$ correspond to a CO observation, that element of $\rho_{\mathbf{o}}^{\mathbf{v}}$ is set to 1. In our initial implementation of EC-CAS, presented
in this work, variable localization is implemented such that meteorological observations do not directly update CO state and CO observations do not update the meteorological state as in Inness et al. (2015). Since Miyazaki et al. (2011) and Kang et al. (2012) show that $CO_2$ updates through wind observations are beneficial, this issue will be considered in future EC-CAS developments. It is worth noting that it is still possible for the CO state to indirectly improve due to the assimilation of wind observations. This can occur because improvement in winds due to meteorological observations leads to an improvement in the





spatial distribution of CO. The spatial correlation between the CO at the location of an observation and that at an unobserved location plays a key role in the update.

While the operational EnKF (Houtekamer et al., 2019) currently uses an ensemble size of 256, EC-CAS uses 64 ensemble members. The main reason for the reduction in ensemble size is to reduce computational cost during the development of EC-CAS. Increases in ensemble size are envisioned in the future. Other EnKF systems used for CO state or flux estimation have used 30 ensemble members (Gaubert et al., 2016; Barré et al., 2015; Miyazaki et al., 2015). In EC-CAS radiosondes, surface stations, ships, aircraft and cloud-drift wind observations are used to constrain the meteorological variables.

The operational EnKF uses an Incremental Analysis Updating (IAU) scheme (Bloom et al., 1996) to control high frequency waves generated by analysis insertion during the ensemble forecasts. This scheme is also applied in EC-CAS to meteorological and GHG analysis increments. To simulate model errors, the operational EnKF uses different model parameters for different ensemble members. These parameters are associated with the most uncertain parameterized physical processes such as boundary layer turbulence and deep convection. Each ensemble member is assigned a unique combination of optional values for ten such parameters. Since the same sources of meteorological forecast error also impact $CO_2$ transport error (i.e. synoptical scale signals (Parazoo et al., 2008; Chan et al., 2004)), the same set of parameters are perturbed in EC-CAS. In the case of CO , the flux estimation results are also sensitive to model errors in convective mass transfer (Jiang et al., 2013, 2011; Ott et al., 2011; Arellano and Hess, 2006) and boundary layer mixing (Arellano and Hess, 2006). The operational EnKF also uses an additive homogeneous, isotropic climatological error produced using the so-called NMC method (Bannister, 2008; Parrish and Derber, 1992) for additional model error simulation. In EC-CAS, for the meteorological assimilation, the same scheme is used, but for GHGs, no such additive error is present. It is not needed for the tests with simulated observations which only seek to characterize system behaviour. However, for the flux estimation extensions (currently under development) an inflation scheme will be implemented. The form of the inflation scheme is yet to be determined. The EnKF code with the extensions developed here is available in Khade et al. (2020).

## 3 Experimental Design

The current paper describes the experiments run for the development and testing of state estimation of CO using simulated observations. The ultimate goal is to develop a system that ingests real GHG observations. However, testing of the system using simulated observations is an important milestone. It not only demonstrates that the system is working properly but also illustrates and defines errors achievable in the best case scenario (under idealized conditions). In this work we use simulated CO observations which are unbiased and have uncorrelated errors.

The experiments in this work are run from 27 December 2014 18:00:00 UTC to 28 February 2015 00:00:00 UTC. At cold start 65 perturbations of meteorological variables are drawn from the same climatological (static) covariance matrix that was

used to generate additive model error. These 65 perturbations are added to the meteorological base state valid at 27 December 2014 18:00:00 UTC. This produces 65 ensemble members for the meteorological variables. Out of the 65 ensemble members the $65^{th}$ ensemble member is designated as the *truth*. The remaining 64 ensemble members are used for the EnKF estimation experiment. With this approach, the truth member is a plausible member of the ensemble having been generated from the same probability density function. The meteorological observations are drawn from the trajectory of the *truth* at 00, 06, 12 and 18Z

every day at which time DA is carried out. The observation networks used in this work are described in section 3.2.

An important facet of this work is accounting for flux error in the CO estimate. This is accomplished through the use of a flux ensemble. The following section 3.1 describes the construction of the flux ensemble while the observations used are discussed in section 3.2.

### 3.1   Flux perturbation

The error in flux is an important source of error in the CO estimate. This is especially true close to the surface. For example, biases in CO analyses near the surface in polluted urban areas were attributed to emissions errors (Inness et al., 2013). Here flux error is simulated using perturbed flux fields. The posterior of the 4D-Var based GEOS-Chem inversion constrained by MOPITT observations is used in this work as the *truth*. These posterior flux fields are constant over a period of one month (see Figure 2).

The flux ensemble, of size 64, is generated by using a spectral algorithm (See Appendix A of Mitchell and Houtekamer (2000)). The flux perturbations are generated such that they are spatially correlated over a distance (half width) of 1000 km. The standard deviation of the spread is set to 40% of the value of the true flux as in Barré et al. (2015). The true flux field is used by the $65^{th}$ ensemble member which generates the truth trajectory. Two sets of 64 flux ensemble members are generated, one each for January and February 2015. Each member of the flux ensemble is used with each (distinct) member of the meteorology

ensemble.

### 3.2   Observation networks

The families of meteorological observations used in this work are summarized in Table 1. The location and times of these observations are *real* though the observation values are *simulated*. These meteorological observations are assimilated in all the experiments presented in this work.

The EnKF is tested with five different CO observational networks. These are summarized in Table 2. The first network, HYPNET is a hypothetical network. It has spatially dense coverage of in situ observations. In this network the observations are located every 1000 km on three planes - 1 km, 5 km and 9 km. These heights are with respect to the local topography. The



| Type | 0000 UTC | 0006 UTC | 0012 UTC | 0018 UTC |
|---|---|---|---|---|
| Upper air | 54765 | 3471 | 51508 | 2057 |
| Aircraft | 78780 | 54065 | 56708 | 78829 |
| Satellite winds | 29749 | 32813 | 33597 | 31719 |
| Surface | 9927 | 10314 | 10271 | 9984 |
| Scatterometer | 18221 | 17484 | 19462 | 16782 |
| GPS-RO | 7064 | 5564 | 6390 | 6527 |

**Table 1.** Columns shows the typical number of meteorological observations assimilated in each 6 h DA cycle.

| Network | Spatial coverage | Temporal coverage |
|---|---|---|
| *HYPNET* | every 1000 km at levels near 1,5,9 km | every 6 hours |
| *ECCC surface* | 17 stations (Canada only) | hourly |
| *GAW surface* | 44 stations | hourly |
| *NOAA surface* | 69 stations | ∼ weekly |
| *MOPITT* | 1 retrieval per 100 km | Global coverage every 3 days. |

**Table 2.** CO observation networks used in this work.

globally averaged topography is 376 m. Therefore, the heights of these planes with respect to mean sea level are roughly 1.376 km, 5.376 km and 9.376 km. The locations of observations are shown in Figure 3a.

The ECCC surface network (Worthy et al., 2005) consists of 17 observing stations in Canada (see Figure 3c, 3d and Table A1). These stations provide measurements at an hourly frequency. Although the ECCC network has expanded rapidly in the past decade to 25 sites in 2020, only the 17 sites providing hourly measurements in 2015 are simulated here. GAW is an acronym of Global Atmospheric Watch (https://gaw.kishou.go.jp). The 44 stations from the GAW network used in the current work are shown in Figure 3c and listed in Table A2. These stations observe at an hourly frequency. The NOAA surface observation net-
work consists of 69 flasks (see Figure 3b). The observations from these flasks are temporally sparse, averaging approximately one per week.

MOPITT (Measurement of pollution in the Troposphere) (Drummond, 1992) is an instrument onboard NASA's Earth Observation Satellite (EOS) Terra that was launched in December 1999. MOPITT is an important component of the global CO observing system because it measures spectra both in the Near InfraRed (NIR) and Thermal InfraRed (TIR) so that its retrieved
profiles are sensitive to CO in the lower troposphere where the flux signal from emissions is most readily detected. That is why it is assimilated in almost all CO data assimilation systems whether their focus is on air quality or flux estimation. It has a nadir footprint of $22 \times 22$ km and a 612 km cross-track scanning swath. Its orbit repeats every 3 days. We used V7J MOPITT data





with locations thinned to one observation per grid box. The coverage on a particular day is shown in Figure 4. The MOPITT averaging kernel is used to construct the MOPITT observation operator. The MOPITT prior denoted by $\mathbf{y}^{pr}$ and the MOPITT

CO retrieval denoted by $\mathbf{y}^{obs}$, are both vectors of dimension 10. Both $\mathbf{y}^{obs}$ and $\mathbf{y}^{pr}$ are defined on 10 levels namely, 1000 hPa, 900 hPa, 800 hPa, ... 100 hPa. In case the surface is not at 1000 hPa, the observations below the surface are ignored.

The averaging kernel, denoted by $\mathbf{A}$, is a matrix of dimension $10 \times 10$. The averaging kernel is the sensitivity of retrieval at each level to all the levels. The observation operator, also known as the forward operator for MOPITT is given by,

$$\mathbf{Hx} \quad = \quad \mathbf{y}^{pr} + \mathbf{A}(\mathbf{x}^{gem} - \mathbf{y}^{pr}) \tag{5}$$

$\mathbf{x}^{gem}$ is the model profile interpolated to the same levels on which the $\mathbf{y}^{pr}$ is defined. The assimilation of MOPITT NIR/TIR retrievals is performed as described in Jiang et al. (2015a).

In the HYPNET, GAW, ECCC and NOAA networks the observation operator is an interpolation operator. The HYPNET, GAW, ECCC and NOAA network are temporally static whereas the MOPITT observations change locations depending on the particular DA cycle. The observation errors are set to 10% of the observation values for all networks in our work. The

validation of the MOPITT V7J data found that the standard deviation of the retrieved profiles varied between 10-16 % relative to independent data (Deeter et al., 2017).

In summary, five different observation networks are simulated out of which four have a significant impact on the CO estimation error. The HYPNET is a hypothetically dense network with uniform spatial coverage and some vertical coverage. This network is useful for identifying coding or other issues with the data assimilation algorithm, since with plentiful, ac-

curate observations and a perfect model a well tuned assimilation scheme should work. The other networks exist in reality and are used to test the assimilation code in a more realistic, but still controlled settings (since no observation biases exist and uncertainty levels are known). Degradation of results relative to the hypothetical network is expected. Our focus is on obtaining a qualitative understanding of the behaviour of the assimilation system in less idealistic settings. It should be noted that although the observations are simulated here, we are not performing Observing System Simulation Experiments (OSSEs)

(See wmo.int/pages/prog/arep/wwrp/new/documents/Final_WWRP_2018_8.pdf for a discussion of designing OSSEs). OSSEs (Prive et al., 2018) are used to compare the results obtained with different observing systems and require careful configuration and tuning of assimilation system parameters so that conclusions might be quantitatively reasonable. Such tuning is difficult with a system that is just being developed. Thus, to reiterate, our results from using different simulated observation networks serve only as a testbed for understanding the new assimilation system.





## 4 Results


This section discusses the improvement in the CO state due to the assimilation of HYPNET, surface observations and MOPITT retrievals, in four seperate experiments. The four CO data assimilation experiments are denoted by *EXP_HYP*, *EXP_GAW*, *EXP_NOAA* and *EXP_MOP*. This improvement is defined with respect to a control experiment which is referred to as *EXP_CNTRL*. *EXP_GAW* assimilates the GAW and ECCC surface observations while *EXP_NOAA* assimilates the NOAA and ECCC surface

observations. This control experiment assimilates simulated meteorological observations (see Table 1) but does not assimilate simulated CO observations. The CO data assimilation experiments assimilate the same meteorological observations as assimilated by *EXP_CNTRL* in addition to their CO observations. The results of the *EXP_CNTRL* are discussed in section 4.1. Section 4.2 illustrates the role of dynamically changing spatial correlations in an EnKF update. An ensemble forecast contains state-dependent correlation information, which to a large extent is dictated by the wind field. In an EnKF this state-dependent

correlation is used to spread observational information to unobserved locations. The results of the four CO data assimilation experiments are described in section 4.3. All experiments are run from 27 December 2014 18:00:00 UTC to 28 February 2015 00:00:00 UTC.

### 4.1 Control experiment

Before delving into the results of the CO data assimilation it is important to examine the results of the meteorological variables.

CO is advected by the winds and hence it is critical to ensure that assimilation of meterological observations is working well. Figure 5 shows the timeseries of area-weighted temperature trial and analysis root mean square error (RMSE) from 27 December 2014 18:00:00 UTC to 28 February 2015 00:00:00 UTC in *EXP_CNTRL* experiment. The RMSE is calculated based on the error between the ensemble mean and the truth which is available at every grid point. As observations are assimilated the RMSE decreases and stabilizes to $0.5°C$ in about 7 days. This timescale is expected since the predictability of weather is about

7-10 days (Pires et al., 1996). After day 7, the errors in synoptic scale motions have saturated at their climatological values. The results for other meteorological variables (not shown) were also examined and it is ascertained that the meteorological data assimilation is working as expected.

The column mean of the ensemble mean of CO averaged over the 7 week assimilation period is shown in Figure 6a. The values of CO are clearly higher in regions of high flux (see Figure 2). The CO from central Africa is advected to equatorial

Atlantic by the easterly winds. Figure 6b shows the ensemble spread of the CO analysis which is estimated by the standard deviation about the analysis mean. This quantifies the EnKF expected error in the CO mean. The spread in the CO ensemble at any grid point is due to perturbations in the flux, spread in the winds and spread obtained by using different realizations of the physics parametrizations within the ensemble and the additive model error term. Figure 6c shows the CO analysis RMSE which quantifies the actual error between the ensemble mean and truth. Clearly, comparing Figure 6a and 6c the RMSE is

higher in regions of higher values of CO. As noted earlier, the regions with high CO correspond to regions of large flux.





The similarity of the spatial pattern of CO ensemble spread (Figure 6b) and the RMSE (Figure 6c) is encouraging because it indicates that the DA system is simulating the actual error well with 64 ensemble members.

The trial and analysis RMSE are identical because CO observations are not assimilated in *EXP_CNTRL*. The time series of RMSE over the period of experimentation is shown by the blue curve in Figure 7a. The RMSE in January 2015 stabilizes to about 16 ppb. The flux field changes in February 2015 and hence the RMSE enters a different regime starting on ∼ 1 February 2015.

The RMSE of the control experiment establishes a baseline against which the RMSEs from CO data assimilations are compared. These comparisons are presented in the section 4.3. The sample error correlations estimated by the ensemble play an important role in ensemble-based data assimilation. This is explained in the next section (4.2).

## 4.2 Role of estimated correlations

The correlations estimated using the trial ensemble plays a key role in spreading the information from a given CO observation to other grid points for any observational network. The correlation estimate changes dynamically depending on the flux perturbations and winds. This state dependence of sample correlation is an important characteristic of ensemble-based filters. The role of the sample correlation and physical localization is illustrated for an observation located at the University of Toronto.

Figure 8 shows the spatial correlation structure for the University of Toronto location at two different times from *EXP_CNTRL*. In equation 1 the term $(\mathbf{y}^o(t) - \mathbf{H}(t)\mathbf{x}^f(t))$ is the *innovation*. This quantity is in the observational space and is thus a scalar for the case of a single observation located at Toronto. Similarly, the matrix inverse in equation 1 is also a scalar. The sample correlation $\mathbf{P}^f(t)\mathbf{H}^T(t)$ is used to map the innovation into model space. This is the correlation between the ensemble of $\mathbf{x}^f(t)$ and $\mathbf{H}(t)\mathbf{x}^f(t)$. For simplicity, we take the nearest gridpoint to Toronto as its actual location so that $\mathbf{H}(t)$ becomes a column vector of the identity matrix with the column index corresponding to the location of the Toronto gridpoint. This renders $\mathbf{P}^f(t)\mathbf{H}^T(t)$ to be the column of $\mathbf{P}^f(t)$ corresponding to the Toronto gridpoint. The scaling factor of the innovation and its uncertainty is neglected since magnitudes are of no concern. The innovation in CO at the University of Toronto location updates the CO at all the other grid points in proportion to the correlation as estimated by the trial ensemble. The regions of high correlation change significantly from 15 January 2015 06:00:00 UTC to 22 January 2015 12:00:00 UTC. Consequently, the impact of the CO observation at University of Toronto on other grid points is different on 15 January 2015 06:00:00 UTC and 22 January 2015 12:00:00 UTC. In theory a given CO observation should update the CO estimate at all other grid points globally. However, due to the physical localization function a given CO observation updates the CO state only within a limited region defined by the horizontal ($hlr$) and vertical localization radius ($vlr$). Localization is necessary for EnKF because the small ensemble size (64) will generate sampling noise in correlations. In other words, small correlations cannot be trusted. Correlations at large physical distance must be filtered because they are most likely spurious and would harm the analysis if retained. The values employed in this work are $hlr = 2000$ km and $vlr = 4$ km. Different values of $hlr$ and $vlr$ were tested and





it was found that the best results were obtained for $hlr = 2000$ km and $vlr = 4$ km. Note that these values of $hlr$ and $vlr$ are used for CO data assimilation only. The assimilation of meteorological observations uses different values for $hlr$ and $vlr$. The red circle in Figure 8 has radius of $hlr = 2000$ km. The Gaspari-Cohn function (Houtekamer and Mitchell, 1998) (not shown in

the Figure) used for physical localization has a peak at University of Toronto and decays moving away from the observation's location. This function is used as a weight to modulate the correlation values. As a result, the impact of the observation decays with distance from the observation. Distance-dependent localization assumes that the sample correlation given by the ensemble is less trustworthy (that is more spurious) as one moves away from the observation. As noted in section 2.4 variable localization ensures that CO observations do not update meteorological variables. Therefore, the estimates of meteorological variables are

the same in all the experiments.

### 4.3  CO DA experiments

*EXP_HYP* assimilates HYPNET observations (see section 3.2) in addition to the same meteorological observations assimilated in *EXP_CNTRL* . The HYPNET observations are assimilated starting on 10 January 2015 18:00:00 UTC after a spin up from 27 December 2014 18:00:00 UTC to 10 January 2015 18:00:00 UTC. This spin up period allows time for the meteorolog-

ical assimilation to stabilize (Figure 5) before the CO data assimilation begins. This spin up also helps the development of correlations within the CO field.

Figure 6d shows the column averaged CO RMSE for the *EXP_HYP* experiment. Compared to the RMSE for the *EXP_CNTRL* experiment the RMSE decreases substantially because HYPNET observations effectively constrain the CO state. The time series of RMSE for the *EXP_HYP* is shown by the red curve in Figure 7a. The blue and red curves overlap from 27 December

2014 18:00:00 UTC to 10 January 2015 18:00:00 UTC during the spin up period. As soon as CO observations are assimilated starting on 10 January 2015 18:00:00 UTC, the RMSE decreases. The reduction in RMSE due to assimilation of HYPNET observations is $\sim 7$ ppb. This reduction is defined as the *benefit*,

$$benefit \quad = \quad RMSE(control) - RMSE(DA) \tag{6}$$

The *relative benefit* is defined as,

$$relative\_benefit \quad = \quad 100 \times \frac{benefit}{RMSE(control)} \tag{7}$$

The second term in equation 6 is the RMSE of the experiment which assimilates CO observations. Since the *EXP_CNTRL* does not assimilate CO observations, *benefit* measures the value of assimilating CO observations from a particular network. This metric quantifies the extent to which CO observations constrain the CO state. Figure 9a shows the spatial structure of



*benefit* in the *EXP_HYP* experiment. This figure is basically the difference between Figures 6c and 6d. The *benefit* is positive

in most parts of the globe except in parts of Tibet and eastern China. A negative value of *benefit* means that assimilation of CO observations increased the RMSE compared to the *EXP_CNTRL*. Negative values can occur because of the statistical nature of data assimilation. However, if the data assimilation system is well tuned, such regions of negative *benefits* should be few and small, as seen here. By comparing the spatial structure of RMSE in Figure 6c and *benefit* in Figure 9a it is clear that the *benefit* is proportional to the RMSE. This makes sense because where the RMSE is large, the observations have a larger scope for

improving the CO state estimate. The relative *benefit* (equation 7) in this experiment is 41%. This means that the assimilation of HYPNET observations decreases the control RMSE by 41%.

The time series of RMSE in *EXP_MOP* is shown by the red curve in Figure 7b. The similarity of the amplitudes of the red curves in Figures 7a and  7b indicates, surprisingly, that the global *benefit* of MOPITT data is only a little worse than that due to the hypotetically dense in situ network (HYPNET). The spatial structure of the *benefit* due to the assimilation of

MOPITT retrievals (*EXP_MOP*) is shown in Figure 9b. Comparing Figure 9a and Figure 9b, it is evident that the *benefit* due to assimilation of HYPNET observations and MOPITT retrievals, is also quite similar in the column mean, in spite of the different spatio-temporal distribution of observations. The relative *benefit* due to assimilation of MOPITT retrievals is 38%.

Figure 10a shows the *benefit* in *EXP_GAW*. Both the GAW and ECCC networks are assimilated in this experiment. The combined network is temporally dense with observations every hour but is spatially sparse except in Canada and western

Europe (Figure 3c and 3d). The relative *benefit* in this experiment is 8%. Figure 10b shows the *benefit* over North America. The bulk of the *benefit* is in the eastern part of this domain though the ECCC stations are located both in eastern and western Canada. The blob of highest *benefit* of about 10-20 ppb is centered on ECCC stations located in Ontario. Though USA does not have any stations in this experiment, the *benefit* of observations in Ontario reaches as far as Florida spreading throughout eastern USA. This is because of the spatial correlation between locations of observations in Ontario and the eastern part of

USA which is evident in Figure 8. The western part of Canada has a weak *benefit* inspite of having several observation sites in this region. This is because the RMSE in the western region is substantially lower than that in the eastern region (see Figure 6c). Thus there is little scope for the observations to improve upon the control RMSE. The relative *benefit* over North America is 38%.

Figure 10a shows that assimilation results in significant *benefit* over Europe and parts of central Africa. The *benefit* in Europe

is due to high spatial density of GAW stations there. However, with only 5 stations in Africa, a *benefit* of 5-20 ppb in central Africa is produced. Some *benefit* is seen over north eastern China and Malaysia due to GAW stations located in these regions.

The last experiment, *EXP_NOAA* assimilates the NOAA flask stations in addition to the ECCC surface stations. It is seen (figure not shown) that *benefit* over Canada is same as that seen in *EXP_GAW*. However, globally the NOAA flask stations do not result in any significant *benefit* over any other region. This is because the flask observations are available, on an average,

only once a week. This experiment is not discussed further in this work.





In the case of the assimilation of HYPNET and MOPITT observations (Figures 9a and 9b) many parts of Atlantic, Pacific, Indian and other oceans show significant *benefit*. The CO flux over oceans is practically zero compared to that on land. The HYPNET and MOPITT observations over oceans contribute to the *benefit* on oceans. However, the improvement of the CO state over land also contributes to be *benefit* over oceans. For example any improvement in CO state over central Africa improves the state over tropical Atlantic ocean due to the downwind transport.

In the discussion so far the horizontal and temporal structure of *benefit* was explored. Figure 11 examines the vertical structure of *benefit*. Figure 11a shows the globally averaged profile of the control RMSE and *benefits* from the three CO DA experiments. The average RMSE of the control experiment (*EXP_CNTRL*) peaks close to the surface with a value of 21 ppb. The average *benefit* in the bottom 4 km is $\sim 7$ ppb in the *EXP_HYP* and *EXP_MOP* experiments, but is only $\sim 1$ ppb for the *EXP_GAW* experiment. Comparing the shapes of the blue and red curve to the black curve, the *benefit* is proportional to the control RMSE except in the bottom $\sim 1$ km. The shape of the *benefit* profile is dictated both by the shape of the control RMSE and the location of observations in the particular network. The *EXP_HYP* profile shows a local peak at 1 km. This is because HYPNET observations are located at 1 km. HYPNET observations are also located at 5 km. However the control RMSE decreases by a factor of 2 from 1 km to 5 km. Consequently the *benefit* also decreases. The peak in the blue profile at $\sim 3$ km is due to a combination of value of RMSE and information content in the MOPITT retrievals.

The profiles averaged over Africa (Figure 11b) have similar shapes to those in Figure 11a. This is because both the RMSE and *benefit* in Africa are high compared to other parts of the globe (see Figures 6c and 9). Hence the global average is dominated by values over Africa. The planes of HYPNET observations are located at roughly 1.3, 5.3 and 9.3 km. The average height of the GAW observations is 2.2 km. The *benefit* in the *EXP_GAW* experiment has a peak value of $\sim 4$ ppb at 3.5 km. This is because the GAW station located at Mount Kenya (0.06° S,37.29° E) has an altitude of 3678 meters. Additionally the site at Assekrem, Algeria located at 23.26° N,5.63° E is situated at an altitude of 2715 meters. The EnKF spreads the information content from these observations to the surrounding regions.

The profiles for North America are shown in Figure 11c. The planes of HYPNET observations are located at approximately 1.3, 5.3 and 9.3 km. The average height of the ECCC observations is 0.38 km. The *benefit* due to the ECCC observations close to the surface is $\sim 4.1$ ppb. The *benefit* in both HYPNET and MOPITT experiments is $\sim 2$ ppb. The *benefit* due to assimilation of ECCC observations decreases monotonically with height because the RMSE decreases monotonically and also because the ECCC observations cannot constrain the CO state beyond the vertical localization radius. The average height of stations in eastern Canada is 212 meters. These stations make a major contribution to the *benefit* over North America. Temporally HYPNET observes every 6 hours while ECCC stations observe every hour. Both, the higher temporal frequency and the lower altitude contribute to the higher *benefit* close to the surface in case of the ECCC observations compared to HYPNET which is located at 1 km.





Figure 11d shows the profiles for Europe. The planes of HYPNET observations are located approximately at 1.19, 5.19 and 9.19 km. The average height of the GAW observations is 1.13 km. In the case of GAW network, the *benefit* is 5.9 ppb close to the surface, The HYPNET *benefit* is much smaller (1.7 ppb). As in the case of North America the better performance of
GAW stations in the bottom 500 meters is a due to both the lower altitude of the stations and higher temporal frequency of observations compared to the HYPNET.

Figure 11e shows the profiles for South America. The planes of HYPNET observations are located at 1.18, 5.18 and 9.18 km. GAW stations are not present in South America and hence the *benefit* is practically zero. The HYPNET and MOPITT *benefits* are approximately proportional to the control RMSE which peaks at 3 km. The South American domain also contains
a large part of the Pacific ocean and some part of Atlantic ocean. The average *benefit* profiles for the *EXP_HYP* and *EXP_MOP* represent the improvement in the oceanic CO state due to the assimilation of observations over the oceans and also the *benefit* due to downwind transport from land regions.

Figures 11f and 11g show an improvement of about 6 ppb close to the surface for *EXP_HYP* and *EXP_MOP* in South and east Asia. The planes of HYPNET observations, in east Asia are located approximately at 1.34, 5.34 and 9.34 km while those in
south Asia are located at 1.45, 5.45 and 9.45 km. The average height of the GAW observations is approximately 0.55 km.

It is evident that the spatio-temporal structure of *benefit* is similar between HYPNET and MOPITT, both horizontally and vertically. This suggests that, in the idealistic setting of unbiased observations and precisely known observation and model error covariances, the performance of MOPITT retrievals is similar to insitu observations. It is should be noted that MOPITT observations are spatially more dense than our HYPNET observation network. In addition, HYPNET has information at three
vertical levels while MOPITT has an information content with one to two degrees of freedom (Deeter et al., 2012) so that vertical information is similar in the two networks.

## 5 Conclusions and further work

A new greenhouse gas data assimilation system based on an operational weather forecast model (EC-CAS v1.0) was developed and validated for the estimation of the 3-dimensional state of CO using simulated observations from HYPNET, ECCC, GAW
and MOPITT networks. The spread in CO is obtained by perturbing the winds, flux fields and physics parametrizations. The CO spread approximately matches the RMSE suggesting that an ensemble size of 64 is acceptable for CO estimation. However, these conclusions are based on the assimilation of simulated observations which are unbiased.

These experiments lead to a qualitative understanding of the decrease in RMSE due to the assimilation of CO observations from realistic networks. With all networks it is seen that the benefit due to assimilation of observations is proportional to the CO
RMSE. Another factor which controls the pattern of benefit is the locations of observations. For example, the GAW network has only one station in central Africa. The observations from this station are able to effectively constrain the CO state within





2000 km. The benefit is the highest in the plane at which this observation is located. The CO state close to the surface is better constrained by observations in the lowermost 500 m than the observations at 1 km. This is suggested by the results in North America and Europe. The CO state over the ocean is constrained partly due to the improvement of the CO state over flux-rich land regions. In the case of MOPITT assimilation, the benefit in central Africa (which is the region with strongest flux) ranges from 10 to over 40 ppb. The downwind transport results in a benefit of 5 to 40 ppb over the tropical Pacific. The benefits over south and east Asia range from 2 to 20 ppb. These quantitative findings are expected to change when real observations are assimilated. Biases in observations and correlations in the observational errors along with unaccounted model errors make assimilation of real observations more challenging so that the error reductions are expected to be smaller.

In the current state of our EnKF, CO prediction (transport) errors due to uncertain meteorological analyses and model formulation errors (e.g. boundary layer transport, convective transport) are encompassed in our forecast ensemble. However, errors in the simplified chemistry model are not addressed. Since we use simulated observations generated by the same model which later performs the CO data assimilation, this type of error is not simulated. However, CO flux estimates are impacted by the choice of OH field (Yin et al., 2015; Jiang et al., 2017, 2015a, b, 2011). On the other hand, Miyazaki et al. (2012) found that in their forecast ensemble, CO flux errors were correlated only with CO fields (in the lower troposphere) and not with other species. In addition, Miyazaki et al. (2015) note that in their multi-species assimilation system, the assimilation of other species had little impact on CO through OH adjustments. This suggests that for flux estimation, simplified CO reactions with OH climatologies may be sufficient, particularly, if the uncertainty in the chemistry can be accounted for. Thus we plan to allow for the uncertainty in the chemistry module through either using an ensemble of OH climatologies or by directly perturbing the reaction rates for CO loss and for the production of CO by methane oxidation.

This work has presented only the very first step in the development of EC-CAS. There are many further stages of development because the goal of EC-CAS is to estimate the 3-dimensional fields of CO, $CO_2$ and $CH_4$ and their fluxes along with meteorological fields by assimilating all available observations of meteorological variables and chemical species using an ensemble smoother. These include both in situ and remotely sensed measurements. The immediate next step is to modify EC-CAS 1.0 to allow the update of the CO flux by CO observations since we demonstrated here that the CO state estimation is working well. Preliminary work suggests that HYPNET observational network is able to estimate the CO flux field after about a week of data assimilation. After the ability to estimate fluxes is demonstrated EC-CAS will be tested for estimation of CO and $CO_2$ 3-dimensional field and their fluxes using real observations. The estimates of flux can be improved by using a smoother rather than a filter since a smoother assimilates future observations too. Ultimately, an ensemble Kalman smoother will be developed.





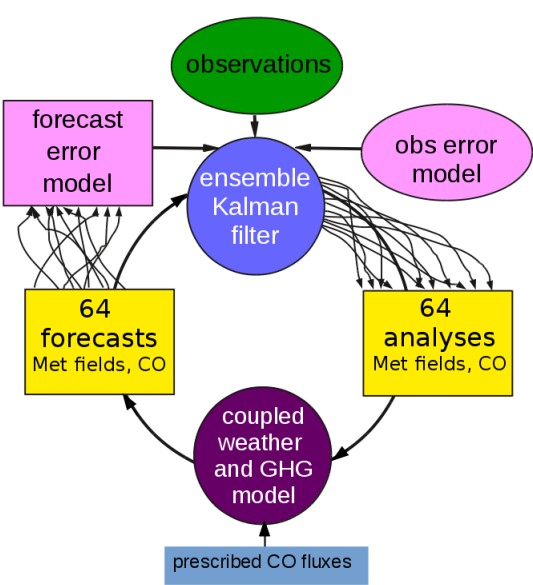

**Figure 1.** EC-CAS v1.0. There are 64 prescribed CO flux ensemble members. In EC-CAS v1.0 flux fields are not estimated. Instead, the perturbed flux field is used to account for uncertainty in CO due to error in the flux. The 64-member forecast ensemble is used along with the observations and the statistics of observation errors as inputs to the EnKF. The 64 analyses of meteorology and CO generated by the EnKF are used as initial conditions for the next 6 hour forecast. This cycle repeats every 6 h.

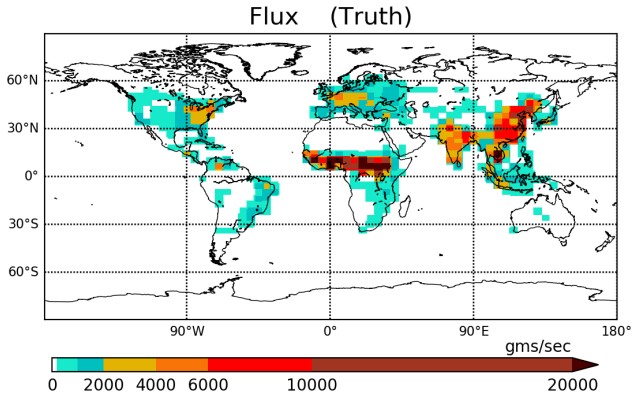

**Figure 2.** The true CO flux field for January 2015.



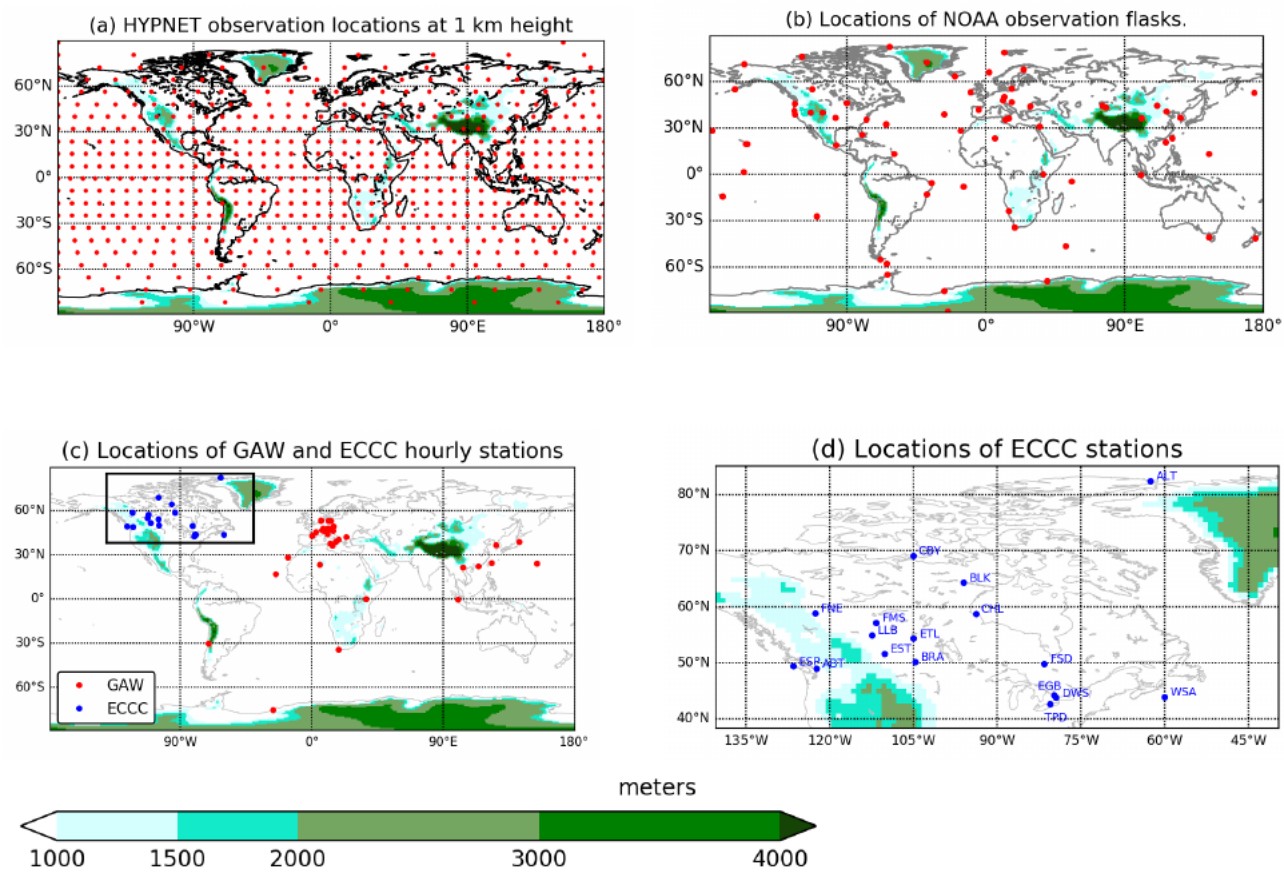

**Figure 3.** In each panel, topography is shown by the color. (a) HYPNET exists at 1 km, 5 km and 9 km. At each of these levels the observations are located 1000 km apart in the horizontal. There are 622 observations locations at each height and total of $622 \times 3 = 1866$ observations every 6 hours. (b) The locations of flask observations from the NOAA network. (c) The GAW and ECCC station locations. (d) The ECCC station locations also shown in panel (c).

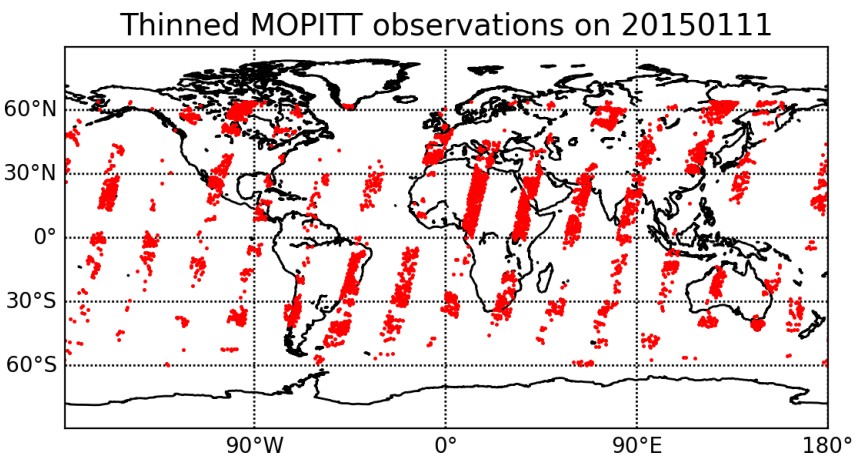

**Figure 4.** An example of the distribution of MOPITT satellite observations. Thinned MOPITT orbits on 11 January 2015 00:00:00 UTC are shown.

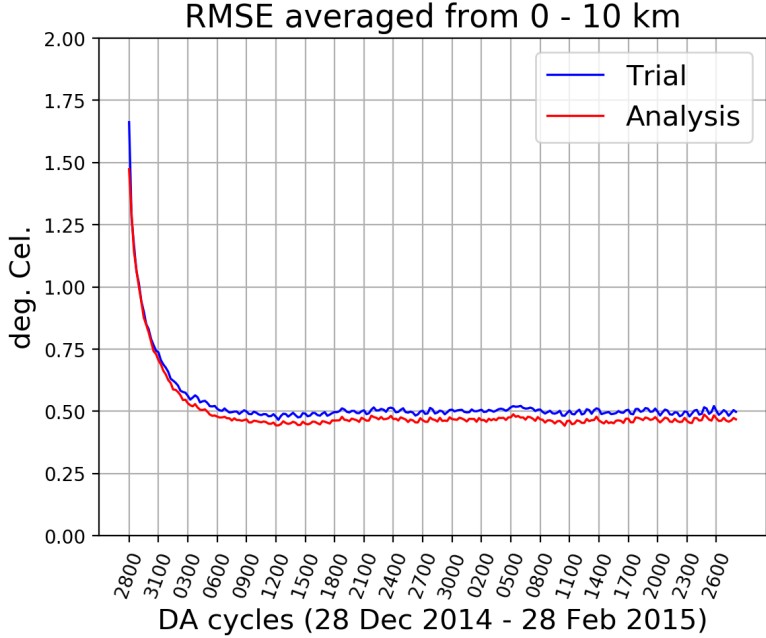

**Figure 5.** Evolution of the root mean squared error of the temperature analyses (red curve) and trial fields (blue curve) from the control experiment which assimilated meteorological observations only.





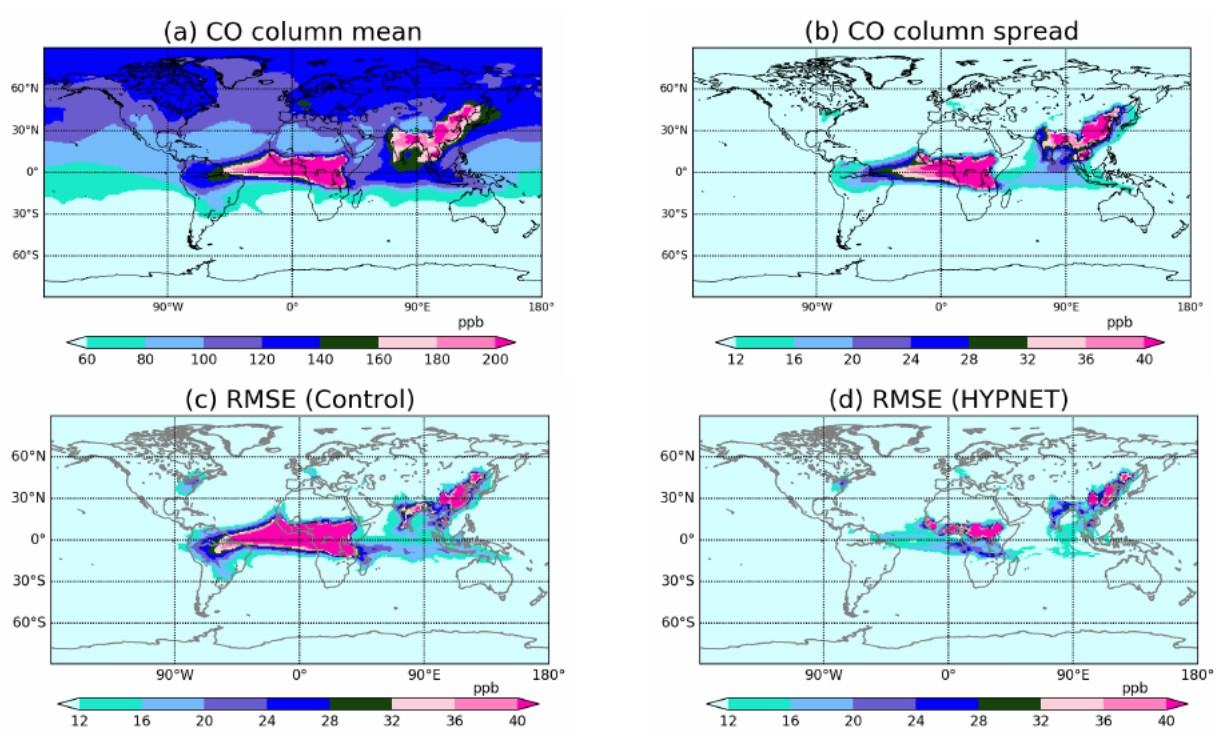

**Figure 6.** In each panel, column means (0-5 km) averaged from 10 January 2015 18:00:00 UTC to 28 February 2015 00:00:00 UTC are shown. (a) CO ensemble mean of the control experiment. (b) CO ensemble spread of the control experiment. (c) RMSE of the control experiment. (d) RMSE of the HYPNET DA experiment.



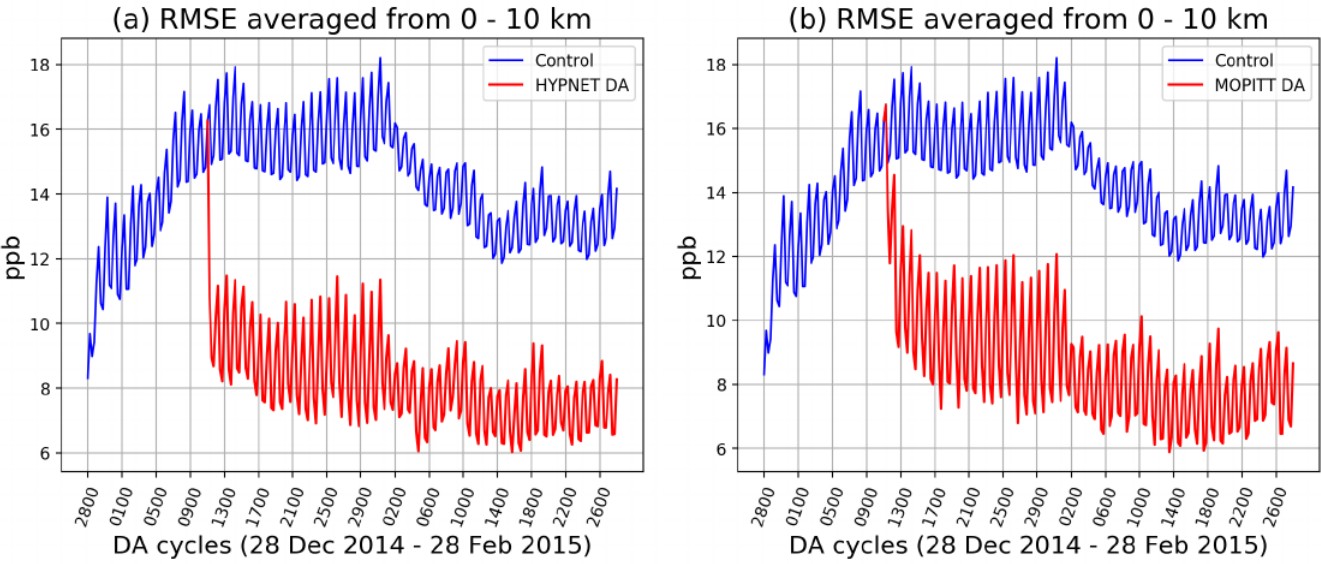

**Figure 7.** Column (0-10 km) mean RMSE of CO analyses from various experiments : (a) The control, *EXP_CNTRL* (blue curve) and the DA experiment assimilating HYPNET observations, *EXP_HYP* (red curve). (b) The control, *EXP_CNTRL* (blue curve) and the DA experiment assimilating MOPITT observations, *EXP_MOP* (red curve). The blue curves in panel(a) and (b) are identical. The 24 h oscillations in the curves are meteorology induced.

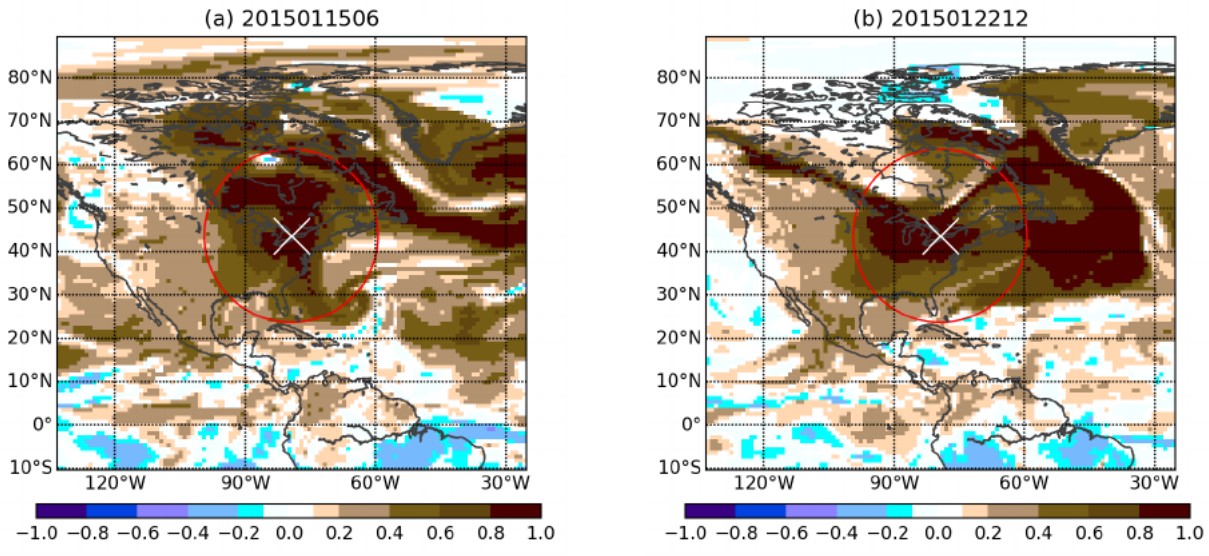

**Figure 8.** The spatial correlation of CO field between University of Toronto (shown by the white cross) and other locations in the horizontal plane defined by the lowest model level. The red circle of radius 2000 km shows the horizontal localization. The correlation is estimated by using the 64 trial ensemble members.



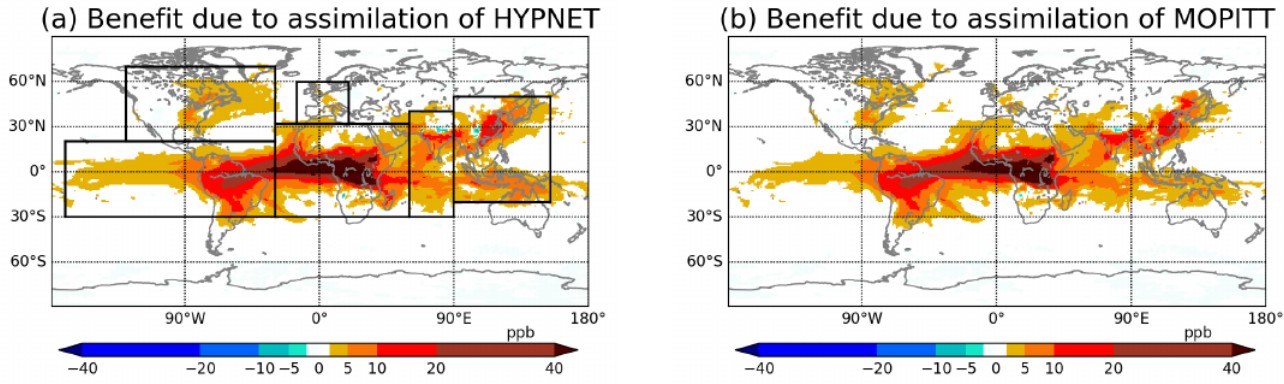

**Figure 9.** Column (0-5 km) mean benefit averaged from 10 January 2015 18:00:00 UTC to 28 February 2015 00:00:00 UTC. In panel (a), the marked boxes show the domains of North America, South America, Europe, Africa, South Asia and East Asia. These domains are used in Figure 11. Note that all the domains contain both ocean and land. The South American domain includes a large part of the Pacific ocean.

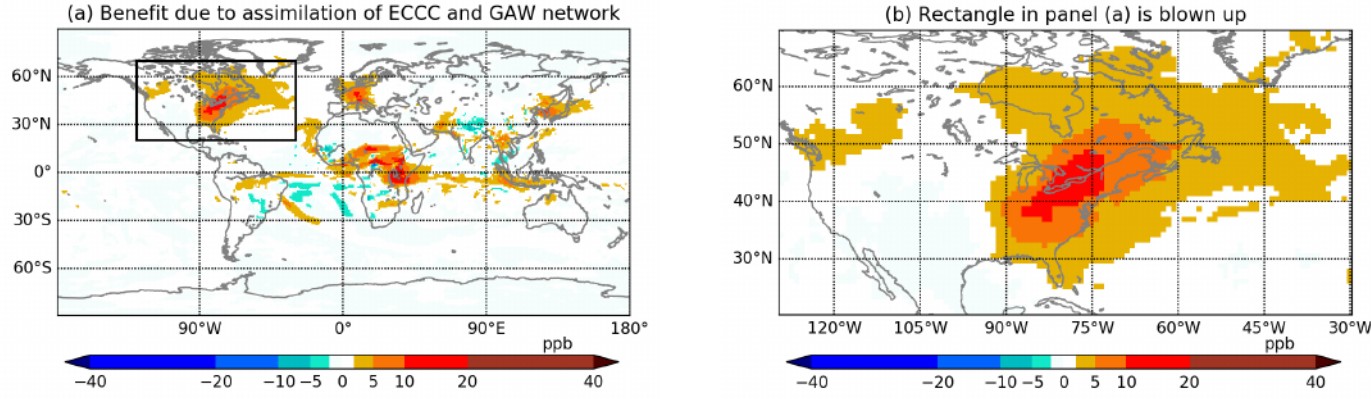

**Figure 10.** (a) Column (0-5 km) mean benefit averaged from 10 January 2015 18:00:00 UTC to 28 February 2015 00:00:00 UTC for the experiment which assimilates near surface measurements from ECCC and GAW networks (*EXP_GAW*). (b) The North American domain from panel (a).





**Figure 11.** Profiles of benefit and control RMSE averaged over different domains. In each panel the black curve shows the control RMSE scaled down by a factor of 3. The colored curves show the benefits due to the assimilation of three different observational networks (*EXP_HYP*, *EXP_MOP* and *EXP_GAW*). Panel (a) shows the globally averaged profiles. The other panels show profiles averaged over domains shown in figure 9a. The limits on the x-axis are different in each panel.





**Appendix**

|    | Code | Station name    | Latitude | Longitude | Altitude |
|----|------|-----------------|----------|-----------|----------|
| 1  | ABT  | Abbotsford      | 49.01    | −122.34   | 60.3     |
| 2  | ALT  | Alert           | 82.45    | −62.52    | 200.0    |
| 3  | BLK  | Baker Lake      | 64.33    | −96.01    | 94.8     |
| 4  | BRA  | Bratts Lake     | 50.20    | −104.71   | 595.0    |
| 5  | CBY  | Cambridge Bay   | 69.13    | −105.06   | 35.0     |
| 6  | CHL  | Churchill       | 58.74    | −93.82    | 29.0     |
| 7  | DWS  | Downsview       | 43.78    | −79.47    | 198.0    |
| 8  | EGB  | Egbert          | 44.23    | −79.78    | 251.0    |
| 9  | ESP  | Estevan Point   | 49.38    | −126.54   | 7.0      |
| 10 | EST  | Esther Alberta  | 51.67    | −110.21   | 707.0    |
| 11 | ETL  | East Trout Lake | 54.35    | −104.99   | 493.0    |
| 12 | FMS  | Fort McKay      | 57.15    | −111.64   | 250.0    |
| 13 | FNE  | Fort Nelson     | 58.84    | −122.57   | 361.0    |
| 14 | FSD  | Fraserdale      | 49.88    | −81.57    | 210.0    |
| 15 | LLB  | Lac LaBiche     | 54.95    | −112.47   | 540.0    |
| 16 | TPD  | Turkey Point    | 42.64    | −80.55    | 231.0    |
| 17 | WSA  | Sable Island    | 43.93    | −60.01    | 5.0      |

**Table A1.** Information about ECCC surface sites used in this study. The altitude is in meters above sea level.



| | Code | Station name | Latitude | Longitude | Altitude | | Code | Station name | Latitude | Longitude | Altitude |
|---|---|---|---|---|---|---|---|---|---|---|---|
| 1 | AMY | Anmyeon-do | 36.5383 | 126.3300 | 71.0 | 23 | IZO | Izaña | 28.3090 | −16.4993 | 2403.0 |
| 2 | ASK | Assekrem | 23.2667 | 5.6333 | 2715.0 | 24 | JFJ | Jungfraujoch | 46.5475 | 7.9851 | 3580.0 |
| 3 | BEO | Moussala | 42.1792 | 23.5856 | 2931.0 | 25 | KMW | Kollumerwaard | 53.3333 | 6.2667 | 3.5 |
| 4 | BKT | Bukit Kototabang | −0.2019 | 100.3180 | 874.0 | 26 | KOS | Kosetice | 49.5833 | 15.0833 | 535.0 |
| 5 | CGR | Capo Granitola | 37.6667 | 12.6500 | 9.0 | 27 | KTB | Kloosterburen | 53.4000 | 6.4200 | 0.0 |
| 6 | CMN | Monte Cimone | 44.1667 | 10.6833 | 2172.0 | 28 | KVV | Krvavec | 46.2973 | 14.5333 | 1750.0 |
| 7 | CPT | Cape Point | −34.3534 | 18.4897 | 260.0 | 29 | LMT | Lamezia Terme | 38.8763 | 16.2322 | 14.0 |
| 8 | CUR | Monte Curcio | 39.3160 | 16.4232 | 1800.9 | 30 | MKN | Mt. Kenya | −0.0622 | 37.2972 | 3682.5 |
| 9 | CVO | Cape Verde | 16.8640 | −24.8675 | 20.0 | 31 | MNM | Minamitorishima | 24.2883 | 153.9833 | 27.1 |
| 10 | ECO | Lecce | 40.3358 | 18.1245 | 86.0 | 32 | NGL | Neuglobsow | 53.1428 | 13.0333 | 62.0 |
| 11 | GAT | Gartow | 53.0657 | 11.4429 | 99.0 | 33 | PAY | Payerne | 46.8129 | 6.9435 | 494.5 |
| 12 | GAT | Gartow | 53.0657 | 11.4429 | 129.0 | 34 | PDI | Pha Din | 21.5731 | 103.5157 | 1478.0 |
| 13 | GAT | Gartow | 53.0657 | 11.4429 | 201.0 | 35 | PDM | Pic du Midi | 42.9372 | 0.1411 | 2881.0 |
| 14 | GAT | Gartow | 53.0657 | 11.4429 | 285.0 | 36 | PUY | Puy de Dome | 45.7723 | 2.9658 | 1467.0 |
| 15 | GAT | Gartow | 53.0657 | 11.4429 | 410.0 | 37 | RIG | Rigi | 47.0674 | 8.4633 | 1036.0 |
| 16 | GLH | Giordan | 36.0700 | 14.2200 | 167.0 | 38 | RYO | Ryori | 39.0319 | 141.8222 | 280.0 |
| 17 | HBA | Halley | −75.3500 | −26.3900 | 38.0 | 39 | SNB | Sonnblick | 12.9578 | 47.0542 | 3111.0 |
| 18 | HKG | Hok Tsui | 22.2095 | 114.2578 | 60.0 | 40 | SSL | Schauinsland | 47.9000 | 7.9167 | 1205.0 |
| 19 | HPB | Hohenpeissenberg | 47.8000 | 11.0200 | 1003.0 | 41 | TLL | El Tololo | −30.1683 | −70.8036 | 2159.0 |
| 20 | HPB | Hohenpeissenberg | 47.8011 | 11.0246 | 1035.0 | 42 | YON | Yonagunijima | 24.4667 | 123.0106 | 50.0 |
| 21 | HPB | Hohenpeissenberg | 47.8011 | 11.0246 | 1078.0 | 43 | ZSF | Zugspitz | 47.4165 | 10.9796 | 2670.0 |
| 22 | HPB | Hohenpeissenberg | 47.8011 | 11.0246 | 1116.0 | 44 | ZUG | Zugspitz-Gipfel | 47.4211 | 10.9859 | 2965.5 |

**Table A2.** Information about GAW surface sites used in this study. The altitude is in meters above sea level.



*Code and data availability.* The source code is publicly available at https://doi.org/10.5281/zenodo.3908545 under the GNU Lesser General Public License version 2.1 (LGPL v2.1) or ECCC's Atmospheric Sciences and Technology licence version 3. The model data output are available at http://crd-data-donnees-rdc.ec.gc.ca/CCMR/pub/2020_Khade_ECCAS_all_data/.

*Author contributions.* Vikram Khade developed the code related to the extension of EnKF to GHGs and their fluxes. Vikram Khade and
Saroja Polavarapu designed, carried out the experiments and interpreted them. Michael Neish prepared the input data required for the experiments. This includes but is not limited to the regridding of flux fields used as the truth. Pieter Houtekamer supervised the development of the code. Seung-Jong Baek helped with code debugging and optimization of the performance of the code. Dylan Jones played an important role in the designing of the experiments and interpretation of the results. Tailong He carried out the 4D-Var inversion using GEOS-Chem to produce flux fields used as truth. Sylvie Gravel helped with the development needed to include $CO$ as a species in the forecast model,
GEM-MACH-GHG.

*Competing interests.* The authors declare that they have no conflict of interest.

*Acknowledgements.* We are grateful to the following people for their advice or assistance. Douglas Chan helped with the understanding of the surface observations. Doug Worthy and his team provided the ECCC surface observations. Feng Deng helped with interpretation of results. Jinwong Kim ran few experiments. Michael Sitwell provided the code to convert CO observations into BURP format. Yves Rochon
provided the MOPITT data and also the code for MOPITT forward operator. Jim Drummond helped with the understanding of the MOPITT data. We thank Michael Sitwell and Yves Rochon for helpful comments on a previous version of the manuscript.



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
