# Peer review of "The Environment and Climate Change Canada Carbon Assimilation System (EC-CAS v1.0): demonstration with simulated CO observations"

_Geoscientific Model Development, 2020_

## Referee Comment (RC1) · Anonymous Referee #1 · 15 Sep 2020

General comments:

The paper describes the new development of the coupled weather and atmospheric composition system based on the Environment and Climate Change Canada's (ECCC's) operational Ensemble Kalman Filter (EnKF). While the paper describes this new configuration as an important advance for the ECCC system it misses important points to provide an accurate and complete description that such system should deserves.

[Figure]

The first major point that needs to be addressed is that the paper advertises in several places that it is a greenhouse gases (GHG) atmospheric data assimilation and surface flux inversion system. However only CO atmospheric data assimilation is showcased. I would strongly recommend that the authors remove all claims that a flux inversion GHG system has been setup and then use a different terminology such as simply "atmospheric composition data assimilation" or "atmospheric carbon data assimilation" as in the title. The study uses synthetic observation to evaluate the system. Therefore, why the authors did not simulate the HYPNET CO2 and CH4 observations and perform the assimilation of such to at least justify the GHG component of the system?

It seems that the added value of the paper is the extension of the ECCC operational system to atmospheric composition using CO assimilation as a proof of concept. While the focus is on CO assimilation, very little importance is given to the meteorology assimilation evaluation in such configuration. How does this compare to the actual operational ECCC system? Almost no references are given to reader to refer to the NWP system and its evaluation. I would recommend the authors to give a short summary on the meteorological data assimilation rather than ascertaining that the meteorological data assimilation is working as expected.

The overall presentation of the paper requires strong efforts to improve clarity. Almost all parts of the paper lack clarity. Some parts are over emphasising some aspects that are not relevant for the evolution of the system while other parts that are important are covered very briefly. To give few examples:

- Very little is explained about the simulation of MOPITT synthetic observations, averaging kernels and their errors. It seems that a paragraph is maybe missing.

- Extensive description of the meteorological setup is given but very little is described and showed about the actual meteorological data assimilation results.

- Some of the terminology used is not really common for atmospheric data assimilation, I would encourage the author to revise this throughout the text.

- Several misleading statements about data assimilation and atmospheric composition need to be corrected. Please refer to the specific comments for details.

Specific comments:

Line 38: Be consistent, so maybe replace by air quality. Or explain that air quality is partly driven by weather.

Lines 39-41: This sentence has some shortcomings that could mislead the reader. Be consistent with the previous sentence and please develop this statement in more precise information. Air quality is a bit different from tropospheric pollution. Tropospheric atmospheric composition prediction is essential to air quality prediction which is looking at surface levels of pollutants. Tropospheric pollution prediction relates to longer time scales than 5 days, especially for CO. Air quality is driven by emissions variations and synoptic variations of weather regimes.

Line 41: Which data assimilation systems are we talking about here?

Line 63 and line 65: Swap years to chronological order

Line 78: This system now can estimate emissions using state augmentation as described in Gaubert et al., 2020 (Gaubert, B., Emmons, L. K., Raeder, K., Tilmes, S., Miyazaki, K., Arellano Jr., A. F., Elguindi, N., Granier, C., Tang, W., Barré, J., Worden, H. M., Buchholz, R. R., Edwards, D. P., Franke, P., Anderson, J. L., Saunois, M., Schroeder, J., Woo, J.-H., Simpson, I. J., Blake, D. R., Meinardi, S., Wennberg, P. O., Crounse, J., Teng, A., Kim, M., Dickerson, R. R., He, H., and Ren, X.: Correcting model biases of CO in East Asia: impact on oxidant distributions during KORUS-AQ, Atmos. Chem. Phys. Discuss., https://doi.org/10.5194/acp-2020-599, in review, 2020.)

Lines 81-82: Maybe this is a bit misleading as the paper seems to focus on CO (even if CO is important for GHG estimations). Also, the term "estimate GHGs" is a bit vague in my opinion. Maybe replace to something more specific such as "estimate CO atmospheric distribution".

Lines 88-91: This paragraph is not necessary here as some of it should be moved to the introduction.

Line 88: "Trial fields" is quite uncommon data assimilation terminology. Maybe replace by forecast, background, prior or first guess fields depending on what you are meaning by trial here.

Lines 88-90: The first and second stage are not explicitly mentioned. I would re-write those two general sentences with a more traditional way to introduce the general concepts of data assimilation.

Lines 94-95: The sentence "The model is initialized. . ." is confusing please rephrase.

Line 95: Please "trial fields" replace with appropriate traditional data assimilation terminology throughout the text.

Line 97: "Blending" is not really the correct word for the data assimilation procedure. I would recommend the author to use the appropriate vocabulary for data assimilation in the literature that tackles atmospheric data assimilation.

Lines 99-101: You do not really need to specify what will be the sections to come here. Consider removing.

Line 108: I do not think that "lib" is the appropriate terminology here. Please again replace with, for example: "... 80 vertical levels from the surface to 0.1 hPa."

Line 109: what type of hybrid coordinate? There are several of them.

Line 114-155: Please be more specific and add diffusivity in this sentence.

Line 120: Not correctly written. The atmospheric chemistry scheme is not removed for CO2. You remove the reactive chemistry in a model to increase its performance. Please rephrase.

Line 130: Start a new paragraph here as you now write about CH4 surface fluxes.

Line 135: Start a new paragraph here as you now write about CO emissions.

Line 148: Please define xf and xa here. xf and xa are commonly called the prior and posterior state respectively in the EnKF terminology. Alternatively, you could call them forecast (hence the superscript f) and analysis (hence the superscript a). Please consider using the commonly used atmospheric data assimilation vocabulary throughout the text for more clarity.

Line 148: Consider directly defining the other elements of the equation 1 before going into explanations.

Line 152-153: The sentence "Pf is the forecast error..." is a bit vague, please be more specific in the definitions.

Line 196: I think there are more relevant papers for this statement. In Inness et al., 2015 the system used was a CTM configuration where the meteorological fields are forced by external meteorological fields. In that sense the DA system could not drive any constrain on the meteorology. Please cite instead Barré et al., 2015 and/or Gaubert et al., 2016 and/or Kang et al., 2012 and so on... Those papers are using EnKF with this variable localisation between atmospheric composition and meteorological variables.

Line 200-201: The sentence "The spatial correlation..." seems to have no link with the previous ones. Please remove or develop in a new paragraph.

Lines 209-210: The sentence "To simulate model ..." is unclear. Please rephrase and possibly add a reference for this error representation method.

Lines 214-215: But this paper is not doing flux estimation. Maybe consider changing to atmospheric composition data assimilation and change to the appropriate references.

Lines 218-219: "In EC-CAS, for the meteorological assimilation, the same scheme is used, but for GHGs, no such additive error is present." Is this the configuration used in this paper? If yes, why then bother going though all these details above?

Line 224: Then why not using synthetic GHG observation of CO2 and CH4 (amongst other GHGs)?

Line 236: change to "the use of a surface flux". I would recommend the authors to be consistent with this terminology as fluxes are not necessarily at the surface in the atmosphere.

Lines 269-270: "... its retrieved profiles are sensitive to CO in the lower troposphere where ..." MOPITT retrievals are sensitive throughout the entire troposphere. The multispectral retrievals allow an enhanced sensitivity towards the surface over land only when the conditions are favourable. Please correct and amend the text accordingly.

Line 271: What are those data assimilation systems? This statement is not true. Number of air quality DAS only assimilate surface stations. Please be more accurate here.

Line 273-274: This statement is misleading. You do not use the averaging kernel to construct the observation operator. You feed the observation operator with the averaging kernel to sample the first guess.

Line 276: This is unclear. Does this mean you discard all observations that have a retrieval surface pressure below 1000 hPa? I do hope you are not doing this. Please clarify the sentence.

Lines 277-278: This is not the proper definition of the averaging kernel matrix. Please use the common definition given by Rodgers 2000.

Inverse Methods for Atmospheric Sounding. Theory and Practice. https://doi.org/10.1142/3171 | July 2000. Pages: 256. By (author):; Clive D Rodgers (Oxford).

Line 278: H is not a forward operator but only an observation operator in the Kalman filter as it does not need to generate a forward model prediction to get a model equivalent quantity. It is true in for example the 4D Var formulation. Please correct.

Line 281: The authors do not use the same system as in Jiang et al., 2015a. If they do this needs to be clearer earlier in the paper. If not, please recall a bit more of the methodology or use the appropriate reference to the system used in this paper.

Line 285: "varied between 10-16%" is this the value that the authors use to set up the observation errors. It seems that few sentences are missing to explain the setup on MOPITT observation errors.

Lines 289-290: This sentence is hard to understand. Please rewrite.

Line 290: "other issues" Please be specific of what other issues.

Lines 293-299: So why do the authors bother simulating observations then? Why not testing the DAS in real conditions? Please justify more clearly the choices here and certainly earlier in the paper.

Lines 308-309: The statement "An ensemble of forecasts..." is incomplete as is, I would remove it as this would need couple sentence to make this point clear and this paragraph is not the place for that.

Lines 311-312: This was already mentioned earlier. Remove.

Line 320: Regarding the reference to Pires et al., 1996, I think numerical weather prediction and predictability ranges have evolved since the mid-90's. Please use a more recent reference. Also, the time of the DAS RMSE stabilisation is not due to weather predictability but mostly due to the DAS setup, i.e. background error, observation density, type and error and so on... Please rewrite the related statements.

Lines 321-322: The authors could add winds, surface pressure and RH (or another NWP variable of your choice) in a four-panel plot to make your point stronger and avoid such statement.

Line 328: What is the "additive model error term"? Is it inflation? Please refer to the section where it is defined and explained? If not define here and/or add the appropriate

reference.

Lines 332-333: This is statistically expected considering Gaussianity and the truth being drawn from the prior distribution itself. Please modify the statement accordingly.

Line 337: change "establishes" by "is"

Line 338-339: The authors should stop recalling what would be the next sub-sections at the end of each sub-sections.

Line 341: Please use more common vocabulary; "trial" is not used in atmospheric data assimilation.

Line 347: what is the matrix inverse. Is it the inverse of P?,H?, R? Or K? Please be more specific.

Lines 351-352: The sentence "The scaling factor. . ." is hard to understand. What is the scaling factor of the innovation? Please define.

Lines 356-357: The statement "In theory a given. . ." is misleading statement. In the theoretical case of a perfect ensemble with an infinite number of members, the spurious correlation would not exist, and you would not need to localise. Hence you would apply the filter globally. Please remove or change accordingly.

Line 359: The statement "small correlations cannot be trusted" is misleading. A GC localisation is not applied to remove small correlations but spurious correlations that are far from the observation location. Small correlations are not necessarily spurious. This also depends on the ensemble size and nature of the state (e.g. lifetime and transport). Please remove or change accordingly.

Line 363: What the meteorological cut off values? Please detail and/or provide reference.

Line 365: Change "has a peak" to "has its maximum".
Line 407: Change "blob" to "area".

Lines 409-410: This is incomplete. The transport of corrected concentration plays a major role as well. I would say this is the combination of both in your case. Please update the text accordingly.

Line 411: If the surface only concentrations and not the 0-5km column were displayed different results might appear as the observation network is at the surface. Also, it is hard to tell in figure 6 that the RMSE is much lower in Western Canada as this is at the edge of the colour scale. The authors should zoom and adjust the plot to make the point clearer.

Lines 441-442: Again, this is not only the EnKF but also the transport of corrected concentration by the model itself that improves the RMSE. Please correct the text.

Line 471: I would disagree with that statement. The vertical information content in the MOPITT retrieval as opposed to HYPNET is not precisely located but spreads across the vertical. So, this is not because the degrees of freedom on the vertical are comparable that the vertical information is similar. Please correct the statement.

Line 473: The authors do not show this as a not directly GHG gas has been assimilated. I would suggest removing GHG but change to something as "atmospheric composition" as only CO assimilation has been demonstrated in the paper.

Lines 484: This is true, but this needs to be reformulated correctly. Please mention atmospheric transport.

Lines 494-495: I am not convinced this is a conclusion from Miyazaki et al., 2012. CO surface flux errors can be correlated with other fields if you consider the co-emission of different species through a given sector. Remove or change the statement accordingly.

Line 506: The authors did not show anything about flux inversions. Please remove this statement.

Line 508: Please define smoother. Add a reference. Use the book from Bocquet et al., 2016 for definitions of the smoother.

Line 509: Future observations? That do not exist unfortunately... Please remove or correct.

Figure 7: What do the numbers in the x-axis indicate? Please clarify or change.

[Figure]

---

## Referee Comment (RC2) · Anonymous Referee #2 · 26 Oct 2020

General comments:

The paper discusses first developments towards an assimilation system for optimizing greenhouse gas concentrations to analyze the carbon cycle globally, but also for Canada. The new developments comprise an extension of the Environment and Climate Change Canada's operationally used Ensemble Kalman Filter to CO observations. The new systems behavior is analyzed using identical twin experiments.
The paper is of general concern for GMD but lack of clarity in the argumentation. While the system aims at analyzing the carbon cycle on a global scale, but also for Canada,

the analyzed time interval from 27 December 2014 to 28 February 2015 is sub-optimal. The papers Fig. 2 suggests that CO fluxes for January 2015 are negligible for Northern Canada. The identical twin experiments should be conducted in the wild fire season to proof the systems ability of analyzing the CO state appropriately even in challenging wild fire episodes. While assimilating synthetic CO observations, the paper claims at several points to be a greenhouse gas assimilation system and that the model state to be optimized is augmented by CO, CO2, and CH4. A clear distinction between future efforts towards the full system, covering also CO2 and CH4, and the current state of the system is not given. In the introduction, it is not made clear how CO assimilation can also improve the concentrations of greenhouse gases. A paragraph about this aspect would be appreciated.

Specific comments:

- generally, the Grammar of the paper, especially the use of commas and articles, should be reviewed

- line 17: replace "GHG" by "greenhouse gases (GHG)"

- line 24: change "2015 or 2016" to "2015 and 2016"

- line 24-25: add a reference for the statement on NIR estimates of anthropogenic CO in Canada

- line 29-34: The claim is not clear. If wild fires are important for the carbon cycle, why not conducting the experiments in the wild fire season. Further, EC-CAS v1.0 does not assimilate CO2 and CH4. Thus, the paper should not claim that EC-CAS does include CO2 and CH4 in the assimilation process.

- line 81: change "GHG and flux estimation" to "CO estimation". In section 2.4 only consider CO estimation in, not GHG. Further, as this is not the purpose of this paper, do not explain details about flux estimation. This should be attributed to the respective paper

- line 95: please add: . . . at every grid point "as well as an perturbed CO emission fluxes."

- line 99 -101: There is no need for repeating the outline of the section. Please remove

- line 107: ". . . and the same lid of 0.1° hPa." Please correct the unit. Do not use "lid", rather use "model top" or equivalent.

- line 116: start a new paragraph

- line 121: replace "This is because. . . is used for..." by "Thus, . . . can be used for"

- line 122: replace: ". . . with an EnKF so the computational expense of complete chemistry is prohibitive and difficult. . ." by "with an EnKF. The computational expense of the complete chemistry would be prohibitive and difficult. . ."

- line 134: add a reference for the statement made in the parenthesis

- line 188, Equation 4: An information about the form of the observation operator, especially for MOPITT-like observations, is missing in the manuscript. Please also consider talking about MOPITT-like observations, rather than MOPITT observations. Further, what is the difference between $\rho_m$ and $\rho_o$?

- line 190: replace "when" by "if"

- line 193: replace "For example when the both the row and column. . ." by "For example, if both, the row and column. . ."

- line 194: replace "that element" by "the respective element"

- line 200-201: This sentence needs to be linked to the rest of the paragraph

- line 213: a table of parameters and the value range would be appreciated

- line 235: Do not start a new paragraph

- line 237-238: Do not include the outline of the next paragraphs

- line 248-250: replace ". . . one each for January and February 2015" by "one for January and February 2015, respectively". Further, rephrase to following phrase.

- line 303: check the line breaking

- line 305: replace "This control experiment assimilates. . ." by "The control experiment (EXP_CNTRL) assimilates. . . "

- line 314: . . . results of assimilating the meteorological variables.

- line 316: the aspect of area-weighted statistics is not made clear. Please provide a description of the weighting procedure.

- line 320: A reference for the climatological values of the temperature uncertainty is missing"

- line 331: . . . RMSE (Figure 6c) and its comparable strength is encouraging. . .

- line 334: . . . over the analysis period is shown by. . .

- line 337-339: Do not include an outline of the next sections

- line 340 (section 4.2): This paragraph lacks on focus. The spatial correlation on two specific days is given. How does this supports the analysis in the subsequent sections? No investigation about the influence of different localization radii is done. The influence of the localization radius on the assimilation results is not investigated. Please consider removing this section or expand it to a more detailed investigation on the localization radii.

- line 371 (section 4.3): throughout this section please be careful in the description of the results. E. g., refer to HYPNET observations but to the EXP_HYP experiment. The same for all other experiments/observation types. This has been mixed up several times.

- line 395: The mean relative benefit. . .

- line 397 – 402: Please make the description more specific by, e. g., including specific values of the benefit.

- line 400: By comparing Fig. 9a and 9b, it is evident that the benefit due to the assimilation of HYPNET observation and MOPITT-like retrievals is also comparable in the column mean, . . .

- line 403: Fig. 10a shows the benefit of the EXP_GAW experiment.

- line 407: replace: "Though USA does not have any stations in this experiment. . ." by "Even though no stations are located in the USA in this experiment, . . ."

- line 414: Fig. 10a shows that assimilation of GAW observations results in . . .

- line 448: please use km as unit for consistency

- line 450: . . . ECCC observations compared to HYPNET observations, which are located at about 1 km.

- Description of Fig. 11: This description is tedious and have to be condensed. For the results of this analysis, the precision of the given height of the observations is irrelevant. Please consider summarizing the mean height information of the different 0station types ad experiments in a table

- line 473: no greenhouse gas assimilation system was presented. Please be more precise

- line 479: .. due to the assimilation of observed CO is proportional...

- line 480: Another factor, which controls the pattern of the benefit, is the location of observations.

- line 482: . . . 2000 km, which is the localization radius used in these experiments.

- line 483: . . . lowermost 500 m than observations at 1 km.

- line 486: replace "Pacific" by "Atlantic"
- Figure 1: change "prescribed CO fluxes" to "ensemble of prescribed CO fluxes"

- Figs. 4 and 5: please increase the resolution of the figures title and axes annotations

- Figure 5: please change the x-axis annotations to dates, same for Fig. 7

- Figure 6: The vertical range of the averaged column (0-5km) is not consistent with other figures, where the range is 0-10 km. Please verify. Further: ... (d) RMSE of the EXP_HYP experiment.

- Figure 8: Spatial correlation of CO between Toronto...

---

## Author Response (AR1)

**Response to Reviewer 1**

The reviewer's comments are in black. Our responses are in blue text. The modifications and additions to the text are highlighted in yellow in the revised manuscript PDF file. However, to see what was deleted, please see the annotated original manuscript PDF file.

**General comments**: The paper describes the new development of the coupled weather and atmospheric composition system based on the Environment and Climate Change Canada's (ECCC's) operational Ensemble Kalman Filter (EnKF). While the paper describes this new configuration as an important advance for the ECCC system it misses important points to provide an accurate and complete description that such system should deserves. The first major point that needs to be addressed is that the paper advertises in several places that it is a greenhouse gases (GHG) atmospheric data assimilation and surface flux inversion system. However only CO atmospheric data assimilation is showcased. I would strongly recommend that the authors remove all claims that a flux inversion GHG system has been setup and then use a different terminology such as simply "atmospheric composition data assimilation" or "atmospheric carbon data assimilation" as in the title. The study uses synthetic observation to evaluate the system. Therefore, why the authors did not simulate the HYPNET CO2 and CH4 observations and perform the assimilation of such to at least justify the GHG component of the system? It seems that the added value of the paper is the extension of the ECCC operational system to atmospheric composition using CO assimilation as a proof of concept. While the focus is on CO assimilation, very little importance is given to the meteorology assimilation evaluation in such configuration. How does this compare to the actual operational ECCC system? Almost no references are given to reader to refer to the NWP system and its evaluation. I would recommend the authors to give a short summary on the meteorological data assimilation rather than ascertaining that the meteorological data assimilation is working as expected. The overall presentation of the paper requires strong efforts to improve clarity. Almost all parts of the paper lack clarity. Some parts are over emphasising some aspects that are not relevant for the evolution of the system while other parts that are important are covered very briefly. To give few examples:

- Very little is explained about the simulation of MOPITT synthetic observations, averaging kernels and their errors. It seems that a paragraph is maybe missing.
- Extensive description of the meteorological setup is given but very little is described and showed about the actual meteorological data assimilation results.
- Some of the terminology used is not really common for atmospheric data assimilation, I would encourage the author to revise this throughout the text.
- Several misleading statements about data assimilation and atmospheric composition need to be corrected.

Please refer to the specific comments for details.

**Response:** We are grateful to the Reviewer for their careful reading of the manuscript and for helpful suggestions. Our original intention was to present our work as the first of a long series of steps to reach our final goal of a greenhouse gas and flux estimation system using an operational weather forecast model. However, both Reviewers felt that the presentation did not sufficiently distinguish the completed work from the context of the desired future work. This led both Reviewers to conclude that the organization of the manuscript was confusing and possibly misleading. We appreciate this feedback and have rewritten the manuscript to focus on only the CO state estimation work which was completed. The overall context and goals of our project are now limited to only the first paragraph in the Introduction, and to a discussion of future work in the Conclusions sections. The term "greenhouse gas" also does not appear anywhere except those two mentioned locations where goals or future work is described. Also, as suggested by the Reviewer, we have provided much more detail about the meteorological assimilation system, both the pre-existing, operational system and our modifications to it in the revised section 2.3. A new supplemental section shows

comparisons of our EC-CAS meteorological estimation with that of the original system (Figures S1-S6) as well as a Table S1 of the types of model perturbations used. Also, the behaviour of meteorological fields in the simulated observation context is now shown with 3 extra panels in Figure 5. In addition, as requested by the Reviewer, the data assimilation terminology was modified as suggested, and all of the specific comments were addressed. Finally, more information about the generation of simulated MOPITT observations was added (line 268-281 in original manuscript and lines 279-297 in the revised manuscript). Overall, we feel that the revised manuscript has greatly benefitted from feedback of both Reviewers. Below, we respond point-by-point to each of the specific comments made by this Reviewer.

**Specific comments:**

Line 38: Be consistent, so maybe replace by air quality. Or explain that air quality is partly driven by weather.
**Response:** Good point. "weather" has been changed to "air quality" (line 38 → line 45 in revised manuscript).

Lines 39-41: This sentence has some shortcomings that could mislead the reader. Be consistent with the previous sentence and please develop this statement in more precise information. Air quality is a bit different from tropospheric pollution. Tropospheric atmospheric composition prediction is essential to air quality prediction which is looking at surface levels of pollutants. Tropospheric pollution prediction relates to longer time scales than 5 days, especially for CO. Air quality is driven by emissions variations and synoptic variations of weather regimes.
**Response:** Thanks for the clarification. "Tropospheric pollution" has been replaced by "Tropospheric atmospheric composition prediction" in this sentence (line 39 → line 45).

Line 41: Which data assimilation systems are we talking about here?
**Response:** The word "those" has been deleted for clarity (line 41 → line 47 ).

Line 63 and line 65: Swap years to chronological order
**Response:** The introduction was rewritten and the paragraph containing these lines was deleted.

Line 78: This system now can estimate emissions using state augmentation as described in Gaubert et al., 2020 (Gaubert, B., Emmons, L. K., Raeder, K., Tilmes, S., Miyazaki, K., Arellano Jr., A. F., Elguindi, N., Granier, C., Tang, W., Barré, J., Worden, H. M., Buchholz, R. R., Edwards, D. P., Franke, P., Anderson, J. L., Saunois, M., Schroeder, J., Woo, J.-H., Simpson, I. J., Blake, D. R., Meinardi, S., Wennberg, P. O., Crounse, J., Teng, A., Kim, M., Dickerson, R. R., He, H., and Ren, X.: Correcting model biases of CO in East Asia: impact on oxidant distributions during KORUS-AQ, Atmos. Chem. Phys. Discuss., https://doi.org/10.5194/acp-2020-599, in review, 2020.)
**Response:** Thanks for the update. The statement on lines 79-80 has been deleted. This reference now appears in the revised section 3.2.

Lines 81-82: Maybe this is a bit misleading as the paper seems to focus on CO (even if CO is important for GHG estimations). Also, the term "estimate GHGs" is a bit vague in my opinion. Maybe replace to something more specific such as "estimate CO atmospheric distribution".
**Response:** "GHGs" has been changed to "CO atmospheric distribution" ( line 82 → line 65).

Lines 88-91: This paragraph is not necessary here as some of it should be moved to the introduction.
**Response:** This paragraph was deleted.

Line 88: "Trial fields" is quite uncommon data assimilation terminology. Maybe replace by forecast, background, prior or first guess fields depending on what you are meaning by trial here.
**Response:** "Trial fields" used to be quite common in atmospheric data assimilation. The problem with "background" is that inverse modellers (especially those dealing with $CO_2$) reserve that term for the large global mean over which local spatial perturbations exist. In addition, inverse modellers use the term "prior" to refer to the flux priors and applying this to the CO state could be confusing. We replaced "trial fields" with "forecast fields" throughout the article. Similarly "trial ensemble" has been replaced by "forecast ensemble" throughout the article.

Lines 88-90: The first and second stage are not explicitly mentioned. I would rewrite those two general sentences with a more traditional way to introduce the general concepts of data assimilation.
**Response:** As noted above, this paragraph was deleted.

Lines 94-95: The sentence "The model is initialized: : :" is confusing please rephrase.
**Response:** The statement has been changed to: "A number (N=64) of 6 h model forecasts are simultaneously integrated from N meteorological and CO initial conditions with forcing from N perturbed CO surface fluxes." (lines 94-95 → 73-75).

Line 95: Please "trial fields" replace with appropriate traditional data assimilation terminology throughout the text.
**Response:** As indicated in the response to comments for Line 88 above, we have replaced "trial fields" throughout the manuscript with "forecast fields". This paragraph (lines 93-101) has been rewritten for clarity (lines 73-83 in revised manuscript).

Line 97: "Blending" is not really the correct word for the data assimilation procedure. I would recommend the author to use the appropriate vocabulary for data assimilation in the literature that tackles atmospheric data assimilation.
**Response:** As noted above lines 93-101 have been rewritten.

Lines 99-101: You do not really need to specify what will be the sections to come here. Consider removing.
**Response:** We have deleted the sentence in lines 99-101.

Line 108: I do not think that "lib" is the appropriate terminology here. Please again replace with, for example: "... 80 vertical levels from the surface to 0.1 hPa."
**Response:** Done. (line 108 → line 89).

Line 109: what type of hybrid coordinate? There are several of them.

**Response:** It is a log hybrid pressure and sigma coordinate that is commonly used (with slight variations) in operational weather forecast models. The provided reference (Girard et al., 2014) describes it in detail.

Line 114-155: Please be more specific and add diffusivity in this sentence.
**Response:** Done. (line 114 → line 95-96).

Line 120: Not correctly written. The atmospheric chemistry scheme is not removed for CO2. You remove the reactive chemistry in a model to increase its performance. Please rephrase.
**Response:** The sentence was changed to: "In contrast, GEM-MACH-GHG uses a simple parameterized chemistry for $CH_4$ and CO while $CO_2$ is treated as a passive tracer." (line 120 → lines 101-102).

Line 130: Start a new paragraph here as you now write about CH4 surface fluxes.
**Response:** Done. (line 130 → line 112).

Line 135: Start a new paragraph here as you now write about CO emissions.
**Response:** Done. (line 136 → line 119).

Line 148: Please define xf and xa here. xf and xa are commonly called the prior and posterior state respectively in the EnKF terminology. Alternatively, you could call them forecast (hence the superscript f) and analysis (hence the superscript a). Please consider using the commonly used atmospheric data assimilation vocabulary throughout the text for more clarity.
**Response:** xf and xa have been defined as the forecast and analysis states. (line 148 → line 145).

Line 148: Consider directly defining the other elements of the equation 1 before going into explanations.
**Response:** Lines 153-154 were moved to lines 146-147 in the revised manuscript.

Line 152-153: The sentence "Pf is the forecast error: : :" is a bit vague, please be more specific in the definitions.
**Response:** We have introduced equations 2 and 3 which define $P^fH^T$ and $HP^fH^T$.

Line 196: I think there are more relevant papers for this statement. In Inness et al., 2015 the system used was a CTM configuration where the meteorological fields are forced by external meteorological fields. In that sense the DA system could not drive any constrain on the meteorology. Please cite instead Barré et al., 2015 and/or Gaubert et al., 2016 and/or Kang et al., 2012 and so on... Those papers are using EnKF with this variable localisation between atmospheric composition and meteorological variables.
**Response:** Actually, Inness et al. 2015 does refer to a coupled system with chemistry modules embedded in the meteorological model. The older MACC system (Inness et al. 2013) was coupled in an offline way to the IFS, but Inness et al. 2015 point out that the earlier approach was not computationally efficient and that chemical tendencies were held fixed for 1 hour and this caused problems at the day/night boundary (see their Introduction). This was the motivation for a fully online chemistry model (called C-IFS). Furthermore, Inness et al. (2015) state on p3 (section 2.2) that "the error covariance matrix between chemical species or between chemical species and dynamics fields is diagonal". Thus variable localization was done by them. However, it is true that other references could be added here. We added Barré et al., 2015 and Gaubert et al. 2016. (line 196 → line 226 ).

Line 200-201: The sentence "The spatial correlation: : :" seems to have no link with the previous ones. Please remove or develop in a new paragraph.
**Response:** The statement was deleted.

Lines 209-210: The sentence "To simulate model : : :" is unclear. Please rephrase and possibly add a reference for this error representation method.
**Response:** The discussion of the meteorological system was moved to the section 2.3 and rewritten. This section now pertains to only the additional changes needed for CO data assimilation.

Lines 214-215: But this paper is not doing flux estimation. Maybe consider changing to atmospheric composition data assimilation and change to the appropriate references.
**Response:** As noted in our response above, this section was rewritten and moved to section 2.3. There is no mention of studies related to flux estimation in the revised section 2.3 and section 2.4.

Lines 218-219: "In EC-CAS, for the meteorological assimilation, the same scheme is used, but for GHGs, no such additive error is present." Is this the configuration used in this paper? If yes, why then bother going though all these details above?
**Response:** This paragraph was rewritten as noted in our last 2 responses. We moved all discussion of the EnKF configuration for meteorology to section 2.3. This then simplifies and clarifies the additional changes needed for CO assimilation.

Line 224: Then why not using synthetic GHG observation of CO2 and CH4 (amongst other GHGs)?
**Response:** Although the model does include $CO_2$ and $CH_4$ as well as CO, and our ultimate goal is to assimilate all 3 constituents, it is a major undertaking to test and validate the system for each of the three species. Each species is quite different in nature and in fact has a completely different literature. Thus the team involved in the validation exercise for a given species would be different. As an example of the differences, $CO_2$ has a lifetime of ~200 years and a very large background with primary surface fluxes from the terrestrial biosphere, ocean, fossil fuel emissions, fires, and land use change. However, $CH_4$ primary surface fluxes include agriculture, wetlands, ocean, anthropogenic emissions, fires and an atmospheric sink and it does not have such a large background value. Thus $CO_2$ has large positive and negative surface sources whereas $CH_4$ has mainly positive surface sources. Thus the best way to simulate surface flux uncertainty in the two cases will differ. Also, because $CO_2$ has a huge background and we are interested in variations of 1-10%, the type of forecast error variance inflation needed will differ from $CH_4$. CO differs again in having a larger dynamic range in mole fractions than either $CO_2$ or $CH_4$ and the shortest lifetime of the three species. Thus, timescales for forecast error variance saturation will differ. Then we come to the observing networks, which are quite different for all 3 species. This is just to name a few of the differences. We do plan to study the assimilation of each species separately, in good time.

Line 236: change to "the use of a surface flux". I would recommend the authors to be consistent with this terminology as fluxes are not necessarily at the surface in the atmosphere.
**Response:** Good point. We now refer to surface fluxes here and elsewhere in the paper.

Lines 269-270: ": : : its retrieved profiles are sensitive to CO in the lower troposphere where : : :" MOPITT retrievals are sensitive throughout the entire troposphere. The multispectral retrievals allow an enhanced sensitivity towards the surface over land only when the conditions are favourable. Please correct and amend the text accordingly.

**Response:** We have revised the sentence to read "retrieved profiles are sensitive to CO in the lower troposphere during daytime over land, where the flux signal from surface emissions is most readily detected." (line 269-270 -→ line 279-281)

Line 271: What are those data assimilation systems? This statement is not true. Number of air quality DAS only assimilate surface stations. Please be more accurate here.

**Response:** We have changed the sentence to read **"**As a result of this sensitivity to lower tropospheric CO, and the long observational record, MOPITT data are widely used for inverse modelling of CO emissions and for air quality studies**"** . We have added references to indicate some of the specific data assimilation systems for which this statement is true. "(e.g. Arelleno and Hess 2006; Fortems et al. 2011; Barré et al. 2015; Jiang et al. 2015b; Yin et al. 2015; Mizzi et al. 2016; Inness et al. 2019; Gaubert et al. 2020;l Miyazaki et al. 2020). (line 271 → lines 281-283 ).

Line 273-274: This statement is misleading. You do not use the averaging kernel to construct the observation operator. You feed the observation operator with the averaging kernel to sample the first guess.

**Response:** We have modified the text to better explain the need to account for the averaging kernel in the observation operator. (Please see lines 288-298 in the revised manuscript).

Line 276: This is unclear. Does this mean you discard all observations that have a retrieval surface pressure below 1000 hPa? I do hope you are not doing this. Please clarify the sentence.

**Response:**The sentence we wrote does not accurately reflect what we did. The 10 levels are a fixed grid. There are no actual observations below the surface. The lowest retrieved level corresponds to the surface level, which may lie at lower pressures than 1000 hPa. We have deleted this sentence and modified line 275 by replacing "1000 hPa" by "surface". (line 275 → line 286).

Lines 277-278: This is not the proper definition of the averaging kernel matrix. Please use the common definition given by Rodgers 2000. Inverse Methods for Atmospheric Sounding. Theory and Practice. https://doi.org/10.1142/3171｜July 2000. Pages: 256. By (author):; Clive D Rodgers (Oxford).

**Response:** Please see the modified text from lines 285-298.

Line 278: H is not a forward operator but only an observation operator in the Kalman filter as it does not need to generate a forward model prediction to get a model equivalent quantity. It is true in for example the 4D Var formulation. Please correct.

**Response:** We have replaced "forward operator" by "observation operator". Please see lines 293-294 in the revised manuscript.

Line 281: The authors do not use the same system as in Jiang et al., 2015a. If they do this needs to be clearer earlier in the paper. If not, please recall a bit more of the methodology or use the appropriate reference to the system used in this paper.
**Response:** The assimilation systems are different, but those differences are not relevant here. We have removed this sentence since explaining the methodology of the Jiang et al. study would not be helpful for the discussion here.

Line 285: "varied between 10-16%" is this the value that the authors use to set up the observation errors. It seems that few sentences are missing to explain the setup on MOPITT observation errors.
**Response:** We have used 10% to set up observations errors. We cite Deeter to justify this value. Please see lines 302-304 in the revised manuscript.

Lines 289-290: This sentence is hard to understand. Please rewrite.
**Response:** We have rewritten these lines.. Please see lines 305-311 in the revised manuscript.

Line 290: "other issues" Please be specific of what other issues.
**Response:** Please see lines 305-311 in the revised manuscript.

Lines 293-299: So why do the authors bother simulating observations then? Why not testing the DAS in real conditions? Please justify more clearly the choices here and certainly earlier in the paper.
**Response:** An important stage in the development of any data assimilation algorithm is to prove that it works. We know from data assimilation theory, that in the absence of bias in observations and models and with plentiful, and accurate observations, the system should work. By simulating observations, we can satisfy the constraints of unbiased observations. We have tried to achieve a balance between a highly idealized setup and reality by allowing the transport model to have imperfections and by using (simulated observations from) real networks like ECCC and MOPITT. Adding different observation networks gives us a further qualitative sense of whether the system is behaving properly since we can guess how using more realistic networks with data gaps will behave relative to the uniformly dense network. Assimilating simulated observations helps us to build confidence in the system we have built. This is only the first step. We will be assimilating real observations. Please see lines 305-311 in the revised manuscript.

Lines 308-309: The statement "An ensemble of forecasts: : :" is incomplete as is, I would remove it as this would need couple sentence to make this point clear and this paragraph is not the place for that.
**Response:** These 2 statements were deleted. We have explained the role of state dependent correlation in spreading observational information in other sections.

Lines 311-312: This was already mentioned earlier. Remove.
**Response:** Done

Line 320: Regarding the reference to Pires et al., 1996, I think numerical weather prediction and predictability ranges have evolved since the mid-90's. Please use a more recent reference. Also, the time of the DAS RMSE stabilisation is not due to weather predictability but mostly due to the DAS setup, i.e. background error, observation density, type and error and so on... Please rewrite the related statements.
**Response:** The statement was deleted.

Lines 321-322: The authors could add winds, surface pressure and RH (or another NWP variable of your choice) in a four-panel plot to make your point stronger and avoid such statement.
**Response:** Figure 5 was expanded to include other meteorological variables.

Line 328: What is the "additive model error term"? Is it inflation? Please refer to the section where it is defined and explained? If not define here and/or add the appropriate reference.
**Response:** The statement has been rewritten for clarity. It now reads: "The spread in the CO ensemble at any grid point is due the perturbations in the flux and those in winds". (lines 328 → 337-338).

Lines 332-333: This is statistically expected considering Gaussianity and the truth being drawn from the prior distribution itself. Please modify the statement accordingly.
**Response:** Though one can make sure that at initial time that the Gaussianity is respected (by drawing from Gaussian flux distribution and initial conditions), one cannot control the extent to which forecast distributions are Gaussian. This is due to the nonlinearities in transport model. Therefore Gaussianity is an assumption that can be violated based on the state (time and location).

Line 337: change "establishes" by "is"
**Response:** Done (line 337 → line 346).

Line 338-339: The authors should stop recalling what would be the next sub-sections at the end of each sub-sections.
**Response:** These lines were deleted.

Line 341: Please use more common vocabulary; "trial" is not used in atmospheric data assimilation.
**Response:** As previously noted, "trial" was replaced by "forecast" throughout the manuscript.

Line 347: what is the matrix inverse. Is it the inverse of P?,H?, R? Or K? Please be more specific.
**Response:** We have rewritten this section to better focus on the illustration of state dependent correlation and localization.

Lines 351-352: The sentence "The scaling factor: : :" is hard to understand. What is the scaling factor of the innovation? Please define.
**Response:** We have rewritten this section. The scaling factor is $(HP^fH+R)$. We have dropped this sentence since it is not central to the illustration of distance dependent localization.

Lines 356-357: The statement "In theory a given: : :" is misleading statement. In the theoretical case of a perfect ensemble with an infinite number of members, the spurious correlation would not exist, and you would not need to localise. Hence you would apply the filter globally. Please remove or change accordingly.
**Response:** "In theory" was meant to indicate exactly this situation of an ensemble with infinite members. However, this is made more explicit in the revised text. (line 356-357 –> 361).

Line 359: The statement "small correlations cannot be trusted" is misleading. A GC localisation is not applied to remove small correlations but spurious correlations that are far from the observation location. Small correlations are not necessarily spurious. This also depends on the ensemble size and nature of the state (e.g. lifetime and transport). Please remove or change accordingly.
**Response:** We agree with the reviewer that both small and large sample correlations can be spurious. We have removed the sentence. However, small correlations are harder to estimate. The sample correlation coefficient has an uncertainty associated with it. The correlation coefficient can be viewed as an estimator. The pdf of this estimator is complex and depends on the sample size and the true correlation (see attached pages from Hoel's Introduction to Mathematical Statistics, Wiley, 1974). For example, with our sample size of 64, for a true correlation of 0.9, the sample correlation coefficient ($r$) will be estimated as lying with the range $0.84 < r < 0.94$ with 95% confidence. However for the same sample size of 64, a true correlation of 0.1 will be estimated as lying between $-0.15 < r < 0.34$ to 95% confidence. Note that the uncertainty range for a high correlation is smaller than that for a low correlation (here a range of 0.1 versus 0.49). So, it is indeed harder to correctly estimate small correlations for a given sample size.

Line 363: What the meteorological cut off values? Please detail and/or provide reference.
**Response:** A reference to the values used is included in section 2.3. Please see lines 186-188 in the revised manuscript.

Line 365: Change "has a peak" to "has its maximum".
**Response:** Done. (line 365 → 366).

Line 407: Change "blob" to "area".
**Response:** Done. (line 407 → 411).

Lines 409-410: This is incomplete. The transport of corrected concentration plays a major role as well. I would say this is the combination of both in your case. Please update the text accordingly.
**Response:** The correlations are a result of the flow-dependent transport. This clarification has been made to the text. We also note the role of downstream transport during the forecasts. Please see lines 414-415 in the revised manuscript.

Line 411: If the surface only concentrations and not the 0-5km column were displayed different results might appear as the observation network is at the surface. Also, it is hard to tell in figure 6 that the RMSE is much lower in Western Canada as this is at the edge of the colour scale. The authors should zoom and adjust the plot to make the point clearer.
**Response:** This is a very good suggestion. We have included two more panels in figure 10. Panel c shows the RMSE and panel d shows the benefit averaged over 0-1 km. The height of the ECCC stations varies from 5 to 707 meters.

Lines 441-442: Again, this is not only the EnKF but also the transport of corrected concentration by the model itself that improves the RMSE. Please correct the text.

**Response:** The EnKF includes the model forecasts as part of the algorithm. However, we believe the Reviewer's point is to distinguish between the analysis step and the forecast step, and indeed both are important for transporting information of observations downstream. (Lines 441-442 → lines 447-448).

Line 471: I would disagree with that statement. The vertical information content in the MOPITT retrieval as opposed to HYPNET is not precisely located but spreads across the vertical. So, this is not because the degrees of freedom on the vertical are comparable that the vertical information is similar. Please correct the statement.

**Response:** The reviewer is correct in noting that the MOPITT information is distributed in the vertical. We have modified the text to state that "HYPNET has information at three vertical levels while MOPITT has an information content with one to two degrees of freedom (Deeter et al., 2012) so that limited vertical information is provided by the two networks." (Lines 471 → lines 464-466).

Line 473: The authors do not show this as a not directly GHG gas has been assimilated. I would suggest removing GHG but change to something as "atmospheric composition" as only CO assimilation has been demonstrated in the paper.

**Response:** We have rewritten the manuscript to avoid discussing GHGs except to mention our future work. We changed "GHG" to "atmospheric composition". (line 473 → 468).

Lines 484: This is true, but this needs to be reformulated correctly. Please mention atmospheric transport.

**Response:** We have modified the statement to include transport. (Line 484 → 480-481).

Lines 494-495: I am not convinced this is a conclusion from Miyazaki et al., 2012. CO surface flux errors can be correlated with other fields if you consider the co-emission of different species through a given sector. Remove or change the statement accordingly.

**Response:** We have deleted this paragraph since it deals with flux estimation and is therefore more relevant to a paper which deals with flux estimation.

Line 506: The authors did not show anything about flux inversions. Please remove this statement.

**Response:** We have removed this statement.

Line 508: Please define smoother. Add a reference. Use the book from Bocquet et al., 2016 for definitions of the smoother.

**Response:** We have defined a smoother. We have added two references – Liebelt for the definition of smoother and Bocquet, 2016 for the formulation of the smoother we want to develop. See lines 494-495 in the revised manuscript.

Line 509: Future observations? That do not exist unfortunately... Please remove or correct.

**Response:** The statement has been clarified to indicate that a smoother uses observations later than the time of the analysis as well as those from earlier than the analysis time. See line 494-495.

Figure 7: What do the numbers in the x-axis indicate? Please clarify or change
**Response:** The x-axis labels refer to the date from 28 Dec. 2014 to 28 Feb 2015. The figure has been revised. Figure 5 has been similarly revised.

**Response to Reviewer 2**

The Reviewer's comments are in black text.  Our responses are in blue text.  The modifications and additions to the text are highlighted in yellow in the revised manuscript PDF file.  However, to see what was deleted, please see the annotated original manuscript PDF file.

**General comments**: The paper discusses first developments towards an assimilation system for optimizing greenhouse gas concentrations to analyze the carbon cycle globally, but also for Canada. The new developments comprise an extension of the Environment and Climate Change Canada's operationally used Ensemble Kalman Filter to CO observations. The new systems behavior is analyzed using identical twin experiments.

The paper is of general concern for GMD but lack of clarity in the argumentation. While the system aims at analyzing the carbon cycle on a global scale, but also for Canada, the analyzed time interval from 27 December 2014 to 28 February 2015 is sub-optimal.  The papers Fig. 2 suggests that CO fluxes for January 2015 are negligible for Northern Canada. The identical twin experiments should be conducted in the wild fire season to proof the systems ability of analyzing the CO state appropriately even in challenging wild fire episodes. While assimilating synthetic CO observations, the paper claims at several points to be a greenhouse gas assimilation system and that the model state to be optimized is augmented by CO, CO2, and CH4. A clear distinction between future efforts towards the full system, covering also CO2 and CH4, and the current state of the system is not given. In the introduction, it is not made clear how CO assimilation can also improve the concentrations of greenhouse gases. A paragraph about this aspect would be appreciated.

**Response:** We thank the Reviewer for the many helpful comments and suggestions.  Both Reviewers felt that the manuscript did not sufficiently distinguish between the completed work (CO state estimation) and the context of our desired future work of building a full greenhouse gas state and flux estimation system. Thus, we have revised the manuscript to mention our context in only the first paragraph of the Introduction, and our future work in the Conclusions section.  The term "greenhouse gas" no longer appears except in these two places.

The focus of this work is on the presentation of the CO state estimation.  Thus, the experiments we performed in winter were adequate for our purposes because there is significant wildfire activity in the tropics to generate reasonable CO fields.  However, for our future flux estimation work, the Reviewer is absolutely correct that we should test the system during the Canadian fire season (boreal summer). Nevertheless, we  carried out some new identical twin experiments in summer 2015. Please see  section 3 in the supplementary material.  Figure S7 compares our true flux fields in January and July 2015.  July 2015 was very active in terms of Canadian forest fires. Therefore the RMSE over North America is much higher in July 2015 than in Jan-Feb 2015. It is seen from the results that the benefit over North America during summer is much larger than that during winter (Figures S8-S10).

We have also provided references to the literature that relate CO and greenhouse gas estimation. Please see revised lines 34-36.  For our system, the initial intention is qualitative: simulations where all three species show the same patterns in a given region indicate a fire source of $CO_2$.  It is also generally expected, by the $CO_2$ flux estimation community, that additional information from other species such as CO will be needed

in the future to attribute surface fluxes to natural or anthropogenic sources. However, the best means for using CO in this fashion has yet to be resolved. Our inclusion of CO prepares for the eventuality of the whole field identifying the best means to constrain $CO_2$ fields with CO measurements.

**Specific comments:**

generally, the Grammar of the paper, especially the use of commas and articles, should be reviewed
**Response :** We have carefully reviewed the manuscript with this focus.

line 17: replace "GHG" by "greenhouse gases (GHG)"
**Response:** Done

line 24: change "2015 or 2016" to "2015 and 2016"
**Response:** Done

line 24-25: add a reference for the statement on NIR estimates of anthropogenic CO in Canada
**Response:** The reference was provided on line 20. It was added again to line 25.

line 29-34: The claim is not clear. If wild fires are important for the carbon cycle, why not conducting the experiments in the wild fire season. Further, EC-CAS v1.0 does not assimilate CO2 and CH4. Thus, the paper should not claim that EC-CAS does include CO2 and CH4 in the assimilation process.
**Response:** This statement speaks about the goal of EC-CAS and our goal is to assimilate all 3 species, though we do only CO here. The forward model does include all 3 species but the assimilation was tested only for CO, so far. We have rewritten the manuscript to avoid mentioning the other species ($CO_2$, and $CH_4$) or surface flux estimation except for the first paragraph which provides motivation and when discussing future plans. In the present work, we are demonstrating that the CO state assimilation is functioning well. Since wildfires occur at all seasons at different locations around the globe, any season is adequate for our present purpose. However, when testing the flux estimation capability it will be necessary to choose the boreal summer when Canadian wildfires tend to occur. We are currently in the process of testing this capability and this extension to our work will be described in a subsequent article. Nevertheless, we also conducted some new experiments for CO state estimation for June 2015 (see the new supplemental material). Indeed, our CO state estimation works just as well in boreal summer when there are larger surface fluxes over Canada from wildfires

line 81: change "GHG and flux estimation" to "CO estimation". In section 2.4 only consider CO estimation in, not GHG. Further, as this is not the purpose of this paper, do not explain details about flux estimation. This should be attributed to the respective paper
**Response:** "GHG" was changed to "CO atmospheric distribution". (Lines 81-82 in original manuscript → lines 64-65 in revised manuscript). Section 2.4 was modified to remove any mention of flux estimation. The title of section 2.4 was changed to "EnKF extensions for CO data assimilation").

line 95: please add: …at every grid point "as well as an perturbed CO emission fluxes."

**Response:** The sentence was rewritten (as requested by Reviewer 1). We also added the point about perturbed fluxes. It now reads: "A number (N=64) of 6 h model forecasts are simultaneously integrated from N meteorological and CO initial conditions with forcing from N perturbed CO surface fluxes". (Lines 94-95 → 73-75 in revised manuscript).

line 99 -101: There is no need for repeating the outline of the section. Please remove

**Response:** Reviewer 1 made the same comment and these lines were deleted.

line 107: "…and the same lid of 0.1° hPa." Please correct the unit. Do not use "lid", rather use "model top" or equivalent.

**Response:** Reviewer 1 had the same point and "lid" was removed. Also the spurious "°" symbol was deleted. . . (line 108 → line 89).

line 116: start a new paragraph

**Response:** Done. See line 98 in the revised manuscript.

line 121: replace "This is because… is used for..." by "Thus, …can be used for"

**Response:** Done. (line 121 → 102-103).

line 122: replace: "…with an EnKF so the computational expense of complete chemistry is prohibitive and difficult…" by "with an EnKF. The computational expense of the complete chemistry would be prohibitive and difficult…"

**Response:** Done. (line 122 → line 104).

line 134: add a reference for the statement made in the parenthesis

**Response:** Our methane simulations have not been published, but we did carry $CH_4$ in the forward model so we had to define a reasonable initial condition because of conversion to CO. This statement describes how this was done.

line 188, Equation 4: An information about the form of the observation operator, especially for MOPITT-like observations, is missing in the manuscript. Please also consider talking about MOPITT-like observations, rather than MOPITT observations. Further, what is the difference between $\rho_m$ and $\rho_o$?

**Response:** The MOPITT observation operator is now better described in section 3.2 . Please see lines 279-298 in revised manuscript. Additionally, we refer to MOPITT-like observations throughout the manuscript. $\rho_m$ and $\rho_o$ are defined immediately after equation 5. See line 177 in the revised manuscript.

line 190: replace "when" by "if"
**Response:** Done.  (line 190 → line 220).

line 193: replace "For example when the both the row and column: : :" by "For example, if both, the row and column…"
**Response:** Done. (Line 193 → Line 223).

line 194: replace "that element" by "the respective element"
**Response:** Done. (Line 194 → line 224).

line 200-201: This sentence needs to be linked to the rest of the paragraph
**Response:** The sentence was deleted as requested by Reviewer 1.

line 213: a table of parameters and the value range would be appreciated
**Response:** A table was added in the supplementary material (Table S1).

line 235: Do not start a new paragraph
**Response:** Done.

line 237-238: Do not include the outline of the next paragraphs
**Response:** Lines 237-8 were deleted.

line 248-250: replace "…one each for January and February 2015" by "one for January and February 2015, respectively". Further, rephrase to following phrase.
**Response:** Done.  (line 248-250 → Line 259).

line 303: check the line breaking
**Response:** Done.

line 305: replace "This control experiment assimilates…" by "The control experiment (EXP_CNTRL) assimilates…"
**Response:** Done.  (line 305 → line 319).

line 314: …results of assimilating the meteorological variables.
**Response:** Done. (line 314 → line 326).

line 316: the aspect of area-weighted statistics is not made clear. Please provide a description of the weighting procedure.
**Response**: This is simply the standard computation in atmospheric modelling for computing global mean quantities on a sphere. The equations are now described in the supplemental material (section 2). Also "area-weighted" was changed to "global mean" in revised line 328.

line 320: A reference for the climatological values of the temperature uncertainty is missing"
**Response:** These sentences were rewritten as per Reviewer 1's request.

line 331: …RMSE (Figure 6c) and its comparable strength is encouraging…
**Response:** Done. (line 331 → lines 341-342).

line 334: …over the analysis period is shown by…
**Response:** Done. (line 334 → line 344).

line 337-339: Do not include an outline of the next sections
**Response:** These lines were deleted as requested by Reviewer 1, also.

line 340 (section 4.2): This paragraph lacks on focus. The spatial correlation on two specific days is given. How does this supports the analysis in the subsequent sections? No investigation about the influence of different localization radii is done. The influence of the localization radius on the assimilation results is not investigated. Please consider removing this section or expand it to a more detailed investigation on the localization radii.
**Response:** This section is meant to be pedagogical. We assume that our paper will be read by researchers who specialize in ensemble techniques but also by researchers who work in inverse techniques but are not familiar with EnKF. This section is meant to illustrate the concept of state dependent covariance estimate and the related issue of localization. Some concepts are easier to explain with the help of pictures in conjunction with equations. In this section we have tried to explain the motivation behind physical localization along with state dependent sample spatial correlation. We did investigate effect of localization radius value on the results and concluded that 2000 km is a good choice. This is mentioned in the revised section 2.4.

line 371 (section 4.3): throughout this section please be careful in the description of the results. E. g., refer to HYPNET observations but to the EXP_HYP experiment. The same for all other experiments/observation types. This has been mixed up several times.
**Response**: We have reviewed the manuscript and ensured the appropriate usage of the terms.

line 395: The mean relative benefit…
**Response:** Done. (line 395 → line 396).

line 397 – 402: Please make the description more specific by, e. g., including specific values of the benefit.
**Response:** We added the values of relative benefit in the last sentence of the paragraph. See lines 405 in the revised manuscript.

line 400: By comparing Fig. 9a and 9b, it is evident that the benefit due to the assimilation of HYPNET observation and MOPITT-like retrievals is also comparable in the column mean,…
**Response:** Done. (line 400 → line 402)

line 403: Fig. 10a shows the benefit of the EXP_GAW experiment.
**Response:** Done. (Line 403 → 406)

line 407: replace: "Though USA does not have any stations in this experiment…" by "Even though no stations are located in the USA in this experiment,…"
**Response:** Done.(line 407-409 → line 412).

line 414: Fig. 10a shows that assimilation of GAW observations results in…
**Response:** Done. (line 414 → line 419).

line 448: please use km as unit for consistency
**Response:** Done (line 448 → line 453).

line 450: …ECCC observations compared to HYPNET observations, which are located at about 1 km.
**Response:** Done. (line 450 → line 455).

Description of Fig. 11: This description is tedious and have to be condensed. For the results of this analysis, the precision of the given height of the observations is irrelevant. Please consider summarizing the mean height information of the different station types ad experiments in a table
**Response:** We agree with the reviewer that broadly speaking the exact height of observations is irrelevant for the analysis. However there are exceptions: eg. the observation at Mt. Kenya spreading observational information in Central Africa and the observations in eastern Canada spreading information to eastern USA. We agree with the reviewer that the discussion is tedious and needs to be condensed. We have shortened the discussion. We have deleted sentences mentioning the heights of HYPNET since it is repetitive. We have retained the discussion about panel (11a) with minor modifications. We decided against including a table. This is because there are qualitative aspects to the connection between heights of observations and the benefit. If a table is included we will have to point the reader to the table for the height and then describe its connection with the benefit. Therefore leaving the heights in the text is better.

line 473: no greenhouse gas assimilation system was presented. Please be more precise
**Response:** "greenhouse gas" was changed to "atmospheric composition". (line 473 → line 468).

line 479: .. due to the assimilation of observed CO is proportional...
**Response:** Done. (line 479 → line 473).

line 480: Another factor, which controls the pattern of the benefit, is the location of observations.
**Response:** Done. (line 480 → line 475).

line 482: …2000 km, which is the localization radius used in these experiments.
**Response:** Done. (line 482 → line 477).

line 483: …lowermost 500 m than observations at 1 km.
**Response:** Done.  (line 483 → 478).

line 486: replace "Pacific" by "Atlantic".
**Response:** Done. (line 486 → line 483).

Figure 1: change "prescribed CO fluxes" to "ensemble of prescribed CO fluxes"
**Response:** Done.

Figs. 4 and 5: please increase the resolution of the figures title and axes annotations
**Response:** We have maximized them.

Figure 5: please change the x-axis annotations to dates, same for Fig. 7
**Response:** Done.

Figure 6: The vertical range of the averaged column (0-5km) is not consistent with other figures, where the range is 0-10 km. Please verify. Further: … (d) RMSE of the EXP_HYP experiment.
**Response:** Figure 7 was changed to show average from 0-5 km.  The Figure caption was corrected.

Figure 8: Spatial correlation of CO between Toronto…
**Response:** Done.

[Figure]

[Figure]

**The Environment and Climate Change Canada Carbon Assimilation System (EC-CAS v1.0) : demonstration with simulated CO observations**

Vikram Khade[1,2], Saroja M. Polavarapu[1], Michael Neish[1], Pieter L. Houtekamer[1], Dylan B.A. Jones[2], Seung-Jong Baek[1], Tailong He[2], and Sylvie Gravel[1]

[1]Environment and Climate Change Canada, 4905 Dufferin Street, Toronto, Canada, M3H 5T4
[2]Department of Physics, University of Toronto, 60 St. George Street, Toronto, Canada, M5S 1A7

**Correspondence:** Vikram Khade (vikram.khade@canada.ca)

**Abstract.** In this study, we present the development of a new coupled weather and greenhouse gas (GHG) data assimilation system based on Environment and Climate Change Canada's (ECCC's) operational Ensemble Kalman Filter (EnKF). The estimated meteorological state is augmented to include three chemical constituents: $CO_2$, CO and $CH_4$. Variable localization is used to prevent the direct update of meteorology by the observations of the constituents and vice versa. Physical localization
5  is used to damp spurious analysis increments far from a given observation. Perturbed flux fields are used to account for the uncertainty in CO due to error in the fluxes. The system is demonstrated for the estimation of 3-dimensional CO states using simulated observations from a variety of networks. First, a hypothetically dense uniformly distributed observation network is used to demonstrate that the system is working. More realistic observation networks based on surface hourly observations, and space-based observations provide a demonstration of the complementarity of the different networks and further confirm the
10  reasonable behaviour of the coupled assimilation system. Having demonstrated the ability to estimate CO distributions, this system will be extended to estimate surface fluxes in the future.

*Copyright statement.* The works published in this journal are distributed under the Creative Commons Attribution 4.0 License. This license does not affect the Crown copyright work, which is reusable under the Open Government Licence (OGL). The Creative Commons Attribution 4.0 Licensea nd the OGL are interoperable and do not conflict with, reduce or limit each other.

15  ©Crown copyright 2020

[revised manuscript text omitted]

A WILEY PUBLICATION

IN MATHEMATICAL STATISTICS

**Introduction to Mathematical Statistics**

**FOURTH EDITION**

**PAUL G. HOEL**

Professor of Mathematics
University of California
Los Angeles

JOHN WILEY & SONS, INC.

New York      London      Sydney      Toronto

[Figure]

Fig. 2. Scatter diagrams and their associated values of $r$.

**2 THE RELIABILITY OF $r$**

If $X$ and $Y$ are jointly normally distributed the pairs of random sample variables $X_i$, $Y_i$, $i = 1, \cdots, n$, will be independently distributed each with the same distribution as $X$ and $Y$. In terms of these random variables the sample means and variances are denoted by $\bar{X}$, $\bar{Y}$, $S_X$, and $S_Y$, and the sample correlation coefficient by

$$r = \frac{\sum_{i=1}^{n}(X_i - \bar{X})(Y_i - \bar{Y})}{nS_XS_Y}.$$

[Figure]

Fig. 3. Distribution for $r$ for $\rho = 0$ and $\rho = .8$ when $n = 9$.

After a sample of size $n$ has been taken and the observational values $(x_1, y_1), \cdots, (x_n, y_n)$ made available, $r$ as given by formula (1) can be computed and is merely a number. However, in discussing how the function $r$ behaves in repeated sampling experiments, $r$ is a random variable which is a function of the $n$ pairs of random variables $X_i$, $Y_i$, $i = 1, \cdots, n$. It is theoretically possible to derive the probability density function of $r$ from the density function of those $n$ pairs of variables; however, both the form and the derivation of this density are too complicated to be considered here. It turns out that the density function of $r$ depends only on the parameters $\rho$ and $n$, where $n$ is the number of points in the scatter diagram. Graphs of the density function of $r$ for $\rho = 0$ and for $\rho = .8$ when $n = 9$ are shown in Fig. 3.

It is clear from Fig. 3 that the distribution of $r$ is decidedly non-normal for large values of $\rho$; consequently it will not suffice to obtain the standard deviation of $r$ and use it to determine the accuracy of $r$ as an estimate of $\rho$. Fortunately, there exists a simple change of variable which transforms the complicated distribution of $r$ into an approximately normal distribution. The resulting normal distribution may then be used to determine the accuracy of $r$ as an estimate of $\rho$ in the same way that the normal distribution of $\bar{X}$ was used to determine the accuracy of $\bar{X}$ as an estimate of $\mu$. This change of variable is from $r$ to $z$, where

(3) $$z = \tfrac{1}{2}\log_e \frac{1+r}{1-r}.$$

It can be shown that when the preceding assumptions are satisfied, the random variable $z$ will be approximately normally distributed with mean

$$\mu_z = \tfrac{1}{2}\log_e \frac{1+\rho}{1-\rho}$$

and standard deviation

$$\sigma_z = \frac{1}{\sqrt{n-3}}.$$

As an illustration of how this transformation is used, consider the problem of determining an interval of values within which $r$ could reasonably be expected to fall if $\rho = .8$ and if $r$ is based on a sample of size 28. The construction of such an interval can be accomplished by first constructing such an interval for $z$ and then transforming it into an interval for $r$. The simplest interval for $z$ that possesses the desired property is the interval with end points $z_1 = \mu_z - 2\sigma_z$ and $z_2 = \mu_z + 2\sigma_z$. For $\rho = .8$ and $n = 28$, it follows from (3) that those end points are

$$z_1 = \tfrac{1}{2}\log 9 - \frac{2}{\sqrt{25}} = .70$$

and

$$z_2 = \tfrac{1}{2}\log 9 + \frac{2}{\sqrt{25}} = 1.50.$$

From tables of the exponential function it will be found that values of $r$ that correspond to these values of $z$ are $r_1 = .60$ and $r_2 = .91$. Thus it can be stated that the probability is approximately .95 that the sample correlation coefficient will satisfy the inequality $.60 < r < .91$ when $r$ is based on a sample of 28 and $\rho = .80$. This example illustrates how unreliable $r$ is as an estimate of $\rho$ unless one has a very large sample.

Although the preceding relationship simplifies the problem of determining the accuracy of $r$ as an estimate of $\rho$, it has the disadvantage of being unreliable if $X$ and $Y$ do not have a joint normal distribution; consequently unless one is quite certain that these variables possess such a distribution, at least to a good approximation, the results should not be relied upon.

**3 INTERPRETATION OF r**

Given any two random variables $X$ and $Y$ one can ask the question whether those variables are independent. Since two variables are independent if, and only if, $f(x, y) = g(x)h(y)$ where $g(x)$ and $h(y)$ are the marginal densities of $X$ and $Y$ and since an extremely large sample would be required to determine whether this relationship is being satisfied, it is clear that some other method is needed to solve this problem. One approach is to introduce some measure of the relationship between two variables, whose value is zero for independent variables, and use it to determine whether the variables are independent.

In view of the definition of $f(x, y)$, it is seen that the normal variables $X$ and $Y$ are independent if, and only if, they are uncorrelated. Thus, the parameter $\rho$ completely determines whether or not two normal variables are independently distributed. As a result, it suffices to determine whether $\rho = 0$ for such a pair of variables to ascertain independence. Since the sample correlation coefficient $r$ serves as an estimate of $\rho$, it can be used to determine whether It is reasonable to assume that $\rho = 0$,

If $X$ and $Y$ cannot be assumed to be normally distributed, even approximately, then $\rho$ can no longer be used as a basis for determining the extent to which $X$ and $Y$ are related. Figure 2 and Fig. 5, Chapter 6, both indicate the inefficiency of $\rho$ and $r$ for measuring the extent of the relationship when $X$ and $Y$ are not normally distributed.

Even though two variables may possess a joint normal distribution, and therefore that $\rho$ may be used as a measure of the strength of the relationship of the two variables, it does not follow that the relationship as measured by $\rho$ is meaningful in a practical sense. The fact that two variables tend to increase or decrease together does not imply that one has any direct or indirect effect on the other. Both may be influenced by other variables in a manner that will give rise to a strong mathematical relationship. The favorite example to illustrate this fact is the one concerned with teachers' salaries. Over a period of years the correlation coefficient between teachers' salaries and the consumption of liquor turned out to be .98. During that period of time there was a steady rise in wages and salaries of all types and a general upward trend of good times. Under such conditions teachers' salaries would also increase. Moreover, the general upward trend in wages and buying power would be reflected in increased purchases of liquor. Thus, the high correlation merely reflected the common effect of the upward trend on the two variables. This is a type of correlation that has received the name of *spurious correlation*. The preceding discussion should make it clear that success with correlation coefficients requires familiarity with the field of application as well as with their mathematical properties and that both the reliability and interpretation of $r$ depend heavily upon the extent to which $X$ and $Y$ are jointly normally distributed.

**4 LINEAR REGRESSION**

As has been observed, empirical correlation methods are often useful in studying how two variables are related. It frequently happens, however, that one studies the relationship between the variables in the hope that any relationship that is discovered can be used to assist in making estimates or predictions of one of the variables. Thus if the two variables are the high